# The Influence of Land Disposition Derived from Land Finance on Urban Innovation in China: Mechanism Discussion and Empirical Evidence

**DOI:** 10.3390/ijerph19063212

**Published:** 2022-03-09

**Authors:** Siyu Han, Mengcheng Wang, Qi Liu, Renyang Wang, Guoliang Ou, Lu Zhang

**Affiliations:** 1College of Public Administration, Huazhong University of Science and Technology, Wuhan 430079, China; siyu@hust.edu.cn (S.H.); wmc@hust.edu.cn (M.W.); liuq2538@hust.edu.cn (Q.L.); 2Research Institute of New Economic, Ningbo University of Finance and Economics, Ningbo 315175, China; wangrenyang@nbufe.edu.cn; 3School of Construction and Environmental Engineering, Shenzhen Polytechnic, Shenzhen 518055, China; 4College of Public Administration, Central China Normal University, Wuhan 430079, China

**Keywords:** land disposition, land finance, urban innovation, dynamic spatial Durbin model, China

## Abstract

As China’s economy advances into a new stage of high-quality development driven by scientific and technological innovation, it is of great practical importance to probe what effects land disposition, which underpinned the previous round of rapid economic growth, and may have an exertion on developing innovation. Based on a deep exploration of the potential positive and negative influences of land disposition in relation to the effects of land finance on urban innovation, we employed a dynamic spatial Durbin model, along with panel data from 266 Chinese prefecture-level cities over the period 2004–2017. The empirical results show that the development of China’s urban innovation has had significant path dependence, spatial agglomeration, and inhibiting effects on neighboring cities, and these effects are attributed to inter-governmental competition and the Matthew effect. Overall, the combined impacts of land disposition modes on urban innovation have changed, from facilitative in the early stage to inhibitory at present. In the developed cities of east China, the facilitative effect of land disposition has weakened gradually, and tends to disappear entirely, while the change in impact over time in less developed mid-western cities is consistent with the national sample. This study broadens our understanding of the role of land disposition in China’s urban innovative development and has meaningful direct implications for policymakers.

## 1. Introduction

Land finance is generally employed by local governments to augment fiscal revenue in China, which is characterized by the heavy reliance of local governments on land-related income (such as land transfer, lease, and tax fees) [1,2,3]. Since the tax-sharing reform in 1994, it has become urgent for local governments to seek extra-budgetary revenue in order to relieve financial pressure as well as promote local development [4,5]. With land finance as an arguably irreplaceable driver, land-centered urbanization and industrialization (yi di mou fa zhan) have both contributed notably to China’s remarkable economic performance in the past few decades [6,7]. Ensuring an abundant supply of industrial land parcels at a lower price in order to attract investments (yi di yin zi), and strategically limiting land provision for commercial and residential use at very high prices in order to collect funds (yi di sheng cai), are two prevalent land disposition modes derived from land finance, and they in turn are of great help to local governments with regard to maximizing their land finance [2,3,8,9,10,11,12,13,14].

Underpinned by land finance, the previous round of China’s economic development has been distinguished by high-speed growth, and is widely considered “a growth miracle” [15,16]. The GDP (Gross Domestic Product) of the whole country had soared from 0.35 × 10^13^ RMB (Renminbi) in 1993 to 10.16 × 10^13^ RMB in 2020, an increase of more than 28.4-fold; China’s share of world GDP was only 1.72% in 1993, and it hit a historic high of about 17% in 2020. However, this high-speed economic growth, driven by natural resources, is unsustainable in the long run due to the limited nature of land resources. Only by transforming such extensive development patterns can the country maintain this momentum in economic growth, and successfully enter the stage of high-quality development [2,17]. Accordingly, the central government developed the Outlines on National Strategy for Innovation-Driven Development in 2016, aimed at developing a new economic growth mode driven by knowledge and technology to preserve competitiveness and realize sustainable development [18]. The report of the 19th congress of the Communist Party of China highlighted that innovation is the primary engine of development, and it is essential to develop new sources of economic growth driven by innovation so that the quality and efficiency of economic development can continue to steadily increase.

It is worth noting that, although the central government has devoted much energy to promoting innovation and inspiring participation in innovation activities, the transition of the economic growth pattern from land-centered to innovation-driven remains difficult [19]. With regard to the root cause, the path dependence of the local development mode and land finance are inseparably intertwined. More specifically, intergovernmental competition has been encouraged by the central government to stimulate economic development by constructing fiscal and political incentives [20]. Even though this governance mechanism has made progress in promoting economic growth, it has also increased the dependence of local governments on land finance, and led to local officials’ blind pursuit of their short-term economic interests and vanity projects. For the sake of risk aversion, innovative projects characterized by high risks, uncertainty, and longer take-off periods would usually not be prioritized in local development strategies [16,19,21]. In part, the lack of support from local governments is a major reason why innovation-driven development is relatively slow in China [2]. In this pivotal stage of economic transformation and upgrading, it is of great practical importance to clarify what effects the land-centered development pattern might have on China’s urban innovation development.

The existing studies that have shed light on this area can be categorized into the following two areas; the first study of the relationship between land finance and land disposition in China. In the context of a GDP-based promotion appraisal system, the land transfer behavior of local governments is aimed at obtaining as much land finance revenue as possible in order to develop the local economy. Thus, two land disposition modes have developed from this heavy dependence on land finance: supplying limited commercial and residential land at higher price by means of bidding, auctions and listings, so as to amass huge capital [4,8,10,22], and leasing out large swathes of industrial land at lower prices through negotiation, so as to attract investments with long-term tax benefits [3,8,10,11,13,23]. However, the tendency to thus seek uneven economic growth that neglects long-term social benefits should not to be overlook [8,14]. Secondly, the number of studies that have discussed how land disposition affects the relevant elements of innovation development has increased in the past few years. One of the most significant effects of this is that local governments can strongly support innovation activities by providing fiscal subsidies, as well as improving infrastructure. However, this is impossible without the high returns generated by commercial and residential land transfers [9,20,24]. Another important positive impact is the agglomeration effect of manufacturing enterprises. In view of the mobility of industrial business, local governments should undertake a low-price transfer strategy with industrial land so as to attract enterprises into urban development zones and industrial parks. The effect of this would be that the industrial structure of a whole city can be upgraded by taking advantage of the technology transfer and knowledge spillover resulting from industrial agglomeration [2,14,25].

Only a small number of scholars have probed the direct influence of land disposition, as derived from land finance, on overall innovation development. Yan et al. found that there is a significant inverted U-curve relationship between land finance and regional innovation at the provincial level—land finance promotes regional innovation at the beginning, however, jeopardizes it in the end [26]. For prefecture-level cities, Xie discovered that land resource misallocation significantly reduces a city’s innovation capacity, and a higher ratio of land occupied by the supply areas of industrial and mining warehouses leads to a lower innovation capacity [19]. Similarly, Xie and Hu verified the comprehensive negative effect of the current land resource allocation approach on urban innovation in China [24]. While the above studies provided some empirical evidence to further explore the mechanisms, by which land disposition influences urban innovation, there are three major limitations. Firstly, although these studies have roughly depicted the way land finance, and the resulting land disposition, affect urban innovation, there is still a limited understanding of all the manifestations and details of the possible influence paths. Second, most of the analyses were carried out from a static perspective, and attempts to track the impacts at different stages of economic growth in China are lacking. Third, the uneven economic development in different regions of China has not been considered, and the regional differentiations in the causes of the impacts of land disposition on urban innovation ought to be taken seriously.

This study aims to deeply investigate the impacts of land disposition as derived from land finance on urban innovation, by incorporating the government-governing mechanism in China. The data cover 266 Chinese prefecture-level cities from 2004 to 2017 were collected as samples, and a spatial econometric model was employed to undertake the empirical analysis. The specific research questions were: (1) How does land disposition derived from land finance influence urban innovation in China? Do spatial spillover effects exist? (2) What changes will take place in the degree of impact as the economy develops in China? (3) What are the similarities and differences in the impact, and how it changes, between developed regions and under-developed regions in China? As compared to the existing studies, the marginal contributions of this study can be summarized as follows: (1) The three aspects of land disposition, land finance and urban innovation have been integrated into a theoretical framework, which facilitates the systematic exploration of the relationship among them; (2) Both the positive and the negative impacts of land disposition on urban innovation have been discussed, and bias may arise if one over-emphasizes the negative effects and neglects the positive; (3) In order to investigate the general features and regional heterogeneity in different areas, an empirical analysis has been undertaken at the national and regional levels, which is helpful in providing relevant implications and policy guidelines for governments that are based on the actual situation.

In the following sections, these questions will be fully analyzed and discussed. The rest of the paper is organized as follows. Section 2 presents the theoretical framework of this study. This is followed by a description of the data in Section 3, and the establishment of the spatial econometric model in Section 4. Section 5 reports the results and provides possible explanations for the findings. The important findings of the study are summarized, and their implications for policy advancement are discussed at the end.

## 2. Theoretical Framework

### 2.1. Institutional Background: How Land Finance and Land Disposition Modes Take Form in China

It is widely accepted that land finance stems from the tax-sharing reform launched in 1994, and the income tax sharing system reform introduced in 2002 [27,28]. The essence of the former reform is a reduction in the local fiscal revenue, along with the decentralization of public expenditure responsibility, while the latter further reduced the main sources of local financial income by changing the status of corporate and personal income tax from a local tax to a central–local shared tax [12,16]. At the same time, though, local governments were allowed to spend their land transfer revenue instead of the central government. Thus, local governments were greatly motivated to pursue land finance revenue in order to escape their fiscal deficits and promote local development [1,29]. Furthermore, the political election system in China, popularly known as yardstick competitions, has greatly increased local governments’ dependence on land finance [8,30]. Intergovernmental competition has played a substantial role in driving China’s economic growth in the context of political centralization and economic decentralization, while the outcomes of competitions determine the promotion prospects of local officials via these yardstick competitions [16,31]. Such a GDP-based performance appraisal system has made economic growth the main administrative duty of local governments [32]. In order to ensure a higher economic growth rate than their rivals, local governments at the same level and under similar economic conditions are often rivals, and imitate each other to improve their promotion prospects, and inevitably, they all end up seeking to maximize land finance revenue within their jurisdiction [4,10,27].

The full opening of China’s urban land transfer market is a critical external condition to the development of land finance. In 2002, the Ministry of Land and Resources issued Decree No.11, which clearly stipulates that urban land transfers for profitable purposes, including commercial, residential, touristic and entertainment, must be carried out publicly in the primary land market by means of tendering, auctioning, and the listing of quotations. Subsequently, the regulation was further extended to the transfer of industrial land in Document No.78, given in 2007. Since local governments have considerable monopolistic and discretionary power over urban land, this regulation has paved the way for them to generate huge land finance revenue by controlling the supply of industrial, residential, and commercial land in the primary land market [30]. The ratio corresponding to the transfer of industrial land to residential and commercial land is 2 in China, while it is just 0.5 in other countries [33]. This phenomenon can be explained by local governments’ mixed land disposition strategies, which they pursue in the name of political gains [19]. Specifically, these involve intervening in the supply quantities and prices of different types of urban land through various transfer means [4,27].

Local governments prefer to strategically provide limited land for commercial and residential use through bidding, auction, and listing, so that market-oriented transfers can increase land value via the competitive pricing mechanism of “the higher bidder gets”. In this way, an enormous one-time influx of land transfer revenue can be obtained immediately [3,10,34]. This revenue is usually employed to construct infrastructure and cover public service costs, so as to meet political performance demands. It is also used to compensate for the fiscal deficits produced by industrial land transfers via agreement-based methods, whereby the low-price strategy of ensuring an abundant supply of industrial land is applied to attract domestic and foreign investment [13,22]. The settlement of manufacturing enterprises can help to rapidly expand the investment scale of fixed assets, and drive immediate local economic growth, meaning investment invitations are one of the most crucial tools used to enhance political performance [28,35]. Since manufacturing enterprises are not location-specific and are very sensitive to costs, local governments become trapped in a “race to the bottom” as regards the transfer policies of their industrial land [9,14,23,36]. They race to offer attractive packages, such as low-priced free land transfers and tax breaks for the first five years, so as to retain investments through closed-door negotiations, in disregard of the market-oriented mechanisms stipulated by the central government [11,12]. Commonly, local governments will investigate investors who have shown an interest in setting up their enterprises locally in advance, and the land transfer price is determined by both the transferor and transferee in private before the land in question is registered in the primary land market. Meanwhile, the enterprise being dealt with will be set as the only bidder, and a transfer through open tender can easily disguise an agreement-based transfer [28,36].

### 2.2. Mechanism Discussion: The Influence Paths of Land Dispositions on Urban Innovation

Against the above-described institutional background, the two land disposition modes of providing limited land for commercial and residential use at high prices through market-oriented transfer methods in order to accumulate capital (LD mode1), and supplying abundant industrial land at low prices by agreement-based transfer methods to attract investments (LD mode2), have gradually taken shape. These two land disposition modes have had profound effects on innovation in many ways, such as via innovation funds, innovation talents, and industrial foundations, which have been proven essential in promoting urban innovation via knowledge production in recent research [24,37,38,39]. A general analytical framework has been established to clearly depict how the above two land disposition modes were derived from land finance that generated positive facilitation pathways and negative inhibition pathways in urban innovation (Figure 1).

In LD mode1, the ever-increasing price of commercial and residential land enables local governments to acquire a sufficient budget to provide financial support that stimulates innovation [2,19]. Theoretically, the key players in innovation activities are manufacturing enterprises, while local governments play the minor role of policy guiders and environment safe-guarders [40]. Driven by the great pressures of political performance assessments and rent-seeking purposes, it is not uncommon for local governments to strongly intervene in the innovation development of manufacturing enterprises by providing a prolonged nanny service, such as by offering adequate subsidies, waiving taxes and fees, assisting in applications for loans, and even direct investment [26]. Since innovation promotion follows the threshold theory, which states that sufficient input is the basis of stable technology outputs, funds provided by local governments can greatly help manufacturing enterprises in accelerating the shift from quantitative to qualitative innovation activities, and eventually enhance their innovation capability. However, along with economic development and innovation improvement, more and more funds from various sources will be invested to help manufacturing enterprises in pursing high-level innovation and reducing their dependence on the funding of local governments [24]. Therefore, the facilitation from land finance revenue to urban innovation may weaken in the long term.

Land finance revenue also contributes considerably to infrastructure financing, as more investments in infrastructure will attract more manufacturing enterprises to settle down [9,20,41]. A lot of the infrastructure construction that is essential for innovation activities would not be possible without the support of local governments, as it is very hard to accomplish with private investment alone, such as information networks, communication equipment, power systems, technological platforms, educational institutions, etc. [24]. The perfection of infrastructure and the construction of industrial development zones have not only helped develop a deeply favorable environment for manufacturing enterprises of higher industrial complexity, but they have also attracted talent in high-tech areas that will help promote productivity [25]. Importantly, there is a Matthew effect in the fight for innovation resources of different cities [26]. Specifically, the combination of innovation funds, innovation talents, and industrial infrastructure plays an active role in the scale expansion and specialization of innovation activities [42]. Furthermore, this agglomeration effect can also cause knowledge spillover between proximate enterprises in the same industry, and help form a denser and more specialized labor market in the city; such resource dividends will then help attract more innovation participators, ultimately forming a positive cycle of growth [26,42]. It is easy to observe this phenomenon in the eastern developed cities of China [24,27]. The advantages of the early accumulation of capital and talent in eastern cities have not only caused traditional manufacturing industries to aggregate, but they have also led to a transfer of labor from inland cities to coastal areas. Thus, the poorer and more remote the cities are, the more difficult it will be to realize effective innovation resource accumulation.

However, LD mode1 also inhibits innovation development in many ways due to the overheating housing market. The over-reliance of local governments on land finance makes housing a seller’s market, and has driven the over-development of the real estate market [2,4]. When compared with innovation projects, real estate businesses can rapidly generate high returns with low risk, so manufacturing enterprises have swarmed towards real estate so as to make robust profits and retain talent. For one thing, manufacturing enterprises often choose to divert some funds from innovation projects to real estate businesses, so that the budget constraints caused by innovation projects can be mitigated to a certain extent, however, such capital distributions often thus have a kind of crowding-out effect on innovation activities [26]. In the long term, the imbalance between the development of real estate businesses and a substantial economy, which results from this crowding-out effect, might trigger a hollowing-out of the manufacturing industry, and consequently impair the capacity for urban innovation [19]. Furthermore, special treatment from the local governments has contributed to the speculative behavior of manufacturing enterprises in the real estate industry. For instance, a semiconductor enterprise in Wuhan applied for residential planning permission on the ground of providing houses at well below market value for its research personnel, while the total area of approved land has far outstripped the actual demands of its qualified staff. Hence, there is no doubt that unoccupied houses will be placed on the housing market sooner or later, and the retention of talent may not be the only reason for manufacturing enterprises to get into real estate.

Skyrocketing housing prices have also increased living costs, and resulted in talent flights. In the early stages of urbanization and industrialization, a suitable position and salary was the first criterion of job location choice for those with innovation talents [24]. However, with the continuous improvement in social and economic factors in certain areas, innovation talents must give priority to house prices and living conditions due to the high living costs when they are offered similar job opportunities in different cities [43,44]. Moreover, the operating costs of producer services industries, such as banking, insurance, law, and other business services, have been unintentionally increased by the tightening of the supply of commercial and residential land. Manufacturing enterprises can focus more on specialized production, since producer services can reduce the transaction costs incurred by a deeper division of labor [45]. Thus, the restrictions imposed on the producer service industry’s development are bound to retard the upgrading of the industrial foundations, which is not conducive to high-level innovation activities.

The positive impacts on urban innovation of LD mode2 can be summarized in three areas. The first is the threshold effect. Such industrial land transfer policies have not only generated innovation demands in the process of industrial development, but have also provided sufficient space to carry out innovation activities. Despite the “race to the bottom” strategy in intergovernmental competitions, land transfer marketization still gives rise to price barriers against external investments, since it will increase the transfer price of industrial land overall [46]. Emerging manufacturing industries with a high degree of complexity and intensity can produce greater value and help local industries to enter into more complex industrial spaces, and will further enhance the upgrade potential of the industrial foundation. With the increase in industrial land transfer prices, traditional industries with low value-added and extensive land use who cannot endure the rising costs would be squeezed out to regions with relatively low land transfer prices [2]. The industrial support policies that depend on labor-intensive manufacturing would also swiftly shift to capital- and technology-intensive industries [23,46]. Such a crowding-out effect on developing manufacturing can speed up the development of urban industrial foundations. The third is the industrial correlation effect. A city with more technologically advanced industrial sectors is more likely to enjoy a favorable economic environment than its less industrialized counterparts, owing to the strong positive spillover effects of intense industrialization on the development of commerce, real estate, and other service industries [14,23,32,47]. Thus, the overall industrial level can be enhanced, and greater levels of innovation talent will aggregate in pursuit of pleasant living conditions [11].

At the same time, LD mode2 has exerted deep adverse impacts on urban innovation. The essence of LD mode2 is the assignment of some of land transfer interest from local governments to manufacturing industries; thus, this mode has contributed significantly in attracting manufacturing assets with mobility. However, the cost of the extensive expansion of a manufacturing enterprise is reduced accordingly, and manufacturing enterprises tend to prefer to increase profits by expanding production scale with the help of a “land grab”, rather than improving traditional production modes through innovation activities [24]. One longer-term result would be a reduction in the innovation incentive, as most manufacturing enterprises are more willing to make steady progress in order to avoid potential risks. Besides this, many cities are ill-equipped to develop technologically advanced industries, and most settled enterprises belong to low-end industries with low skill requirements and market competitivity [33]. Diversity theory suggests that the diversity of productive activities in a city drives the promotion of innovation; it is thus very hard for homogeneous under-developed manufacturing industries to break through the barriers preventing their innovation activities, and this thereby limits their industrial transformation and technological innovation [33,43,46].

Land development across the country has deeply strengthened the administrative interventions into the land market, and undermined the basic role of market forces in allocating land resources. Land deals dominated by guanxi have provided breeding grounds for corruption, and caused more land violations [10]. The phenomenon of rent-seeking and collusion between local cadres and their political coalitions will inhibit innovation eventually as manufacturing enterprises tend to improve firm performance by acquiring cheaper land, instead of improving their own innovation abilities under such institutional conditions [10,19]. This is the crowding-out effect on innovation activities, as caused by administrative intervention.

The lack of attraction for those with innovation talents is another important negative impact of the LD2 mode. Most industry parks are far from the city center and thinly populated, meaning the commerce and leisure demands of those individuals cannot be satisfied as some service industries will not be provided. Poor living conditions have become a common cause of talent flight in some cities [3]. Moreover, the excessive extension of industrial land may cause pollution, which will not only affect the innovation vitality of highly talented individuals through physical and emotional illness, but also reduce the region’s innovation vitality [5,44,48]. All in all, a good urban living environment is crucial to innovation talent, and the overall innovation level of a city.

## 3. Materials and Methods

### 3.1. Variable Selection and Description

Three types of variables were selected to develop the quantitative relationship between land disposition and urban innovation, including the proxy variables of dependent variables, independent variables, and control variables.

As the dependent variable, the urban innovation index from the FIND Report on City and Industrial Innovation in China (2017) was selected to quantify the overall level of urban innovation [49]. We also calculated the value of urban innovation in the following year using the growth rate to enlarge the sample. Based on the patent data and the micro–big data of the China Intellectual Property Office, and the registered capital data of the China Industrial and Commercial Bureau, the urban innovation index was calculated using the patent update model developed by Kou and Liu. It is worth mentioning that although many studies employed the number of patents applied or authorized to measure the innovation level of a city [43,50,51,52,53], there are limitations to this approach. This is due to statistical errors being inevitable when searching patent information manually, and it is hard to infer the true value of urban innovation by only considering the number of patents.

As the independent variable, two variables were selected to stand as proxy variables for LD mode1 and LD mode2; they were the price departure of market-oriented land transfer with agreement-based land transfer, and the proportion of the total area of agreement-based land transfers, respectively. Under LD mode1, local governments raise the price of commercial and residential land in the open competitive land market to compensate for the fiscal deficit resulting from the low-yielding strategy of industrial land transfers through agreements [23]. In LD mode2, local governments transfer more land into industrial sectors in order to attract investment and promote economic growth [54]. These different land transfer behaviors lead to differences in the supply structure between commercial and residential land and industrial land, and thus further generate land transfer price differences and land resource misallocations [19].

For the control variable, a group of indicators related to urban innovation were screened. (1) Economic development level. The marketization level increases with the development of the urban economy, and an active market economy would more effectively drive innovation investment. (2) Investment scale of Research and Development (R&D). This can determine the level of support via innovation funding given by local governments, and strongly influences the development of urban innovation. (3) Higher education level. The development of higher education leads to the expansion of high-quality talent capital, and industry–university–research institution cooperation has a beneficial stimulating effect on innovation activities. (4) Degree of the opening-up. An increase in opening-up is conductive to promoting industrial development and further improving the urban innovation level [2]. (5) Financial development level. Innovation projects often face financial restrictions due to the nature of venture activities, while financial support from national banking systems can to some extent relieve innovation funding pressures [19]. The details of these variables are shown in Table 1.

### 3.2. Empirical Approach

#### 3.2.1. Global Moran’s I Index

The innovation development of one city may show similar variation tendencies to neighboring cities due to the spillover and diffusion effects; such correlation between neighboring spatial units is called the spatial correlation [55]. Constructing a spatial econometric model is the main approach to investigating spatial correlation in some of the previous studies [8,13,30,56]. However, it is first necessary to examine whether the dependent variable has spatial dependence before conducting spatial economic analysis. The Global Moran’s I index is the most widely used method to measure whether a spatial correlation exists, and this reflects the degree of similarity between the attribute values of spatially adjacent or neighboring areas. The value is range (−1, 1). Greater than zero means the given variable shows positive spatial connections among its observations and the existence of cluster in the spatial arrangement. Smaller than zero indicates that the given variable has negative spatial connections. When the value is zero, there is no spatial connection among the observation variables, and a random distribution in space can be inferred [57,58]. The calculation formula is as follows:(1)$$\mathrm{Mora}n^{\prime }s\;I=\frac{\sum_{i=1}^{n}\sum_{j=1}^{n}W_{ij}\left(Y_{i}-\bar{Y}\right)\left(Y_{j}-\bar{Y}\right)}{S^{2}\sum_{i=1}^{n}\sum_{j=1}^{n}W_{ij}},\;\;S^{2}=\frac{1}{n}\overset{n}{\underset{i=1}{\sum }}{\left(Y_{i}-\bar{Y}\right)}^{2}$$

In Equation (1), $Y_{i}$ and $Y_{j}$ denote the urban innovation level of i city and j city, respectively; n is the number of cities; $W_{ij}$ is the spatial weight matrix; $\bar{Y}$ is the mean value of the samples; and $S^{2}$ is the variance of the samples. In this study, a binary contiguity spatial weight matrix was constructed to reflect spatial adjacency. When region i is adjacent to region j, $W_{ij}$ = 1; when region i is not adjacent to region j, $W_{ij}$ = 0.

#### 3.2.2. Spatial Econometric Model

According to the First Law of Geography, everything is related to everything, and closer things correlate more closely than distant ones; therefore, biased results may be generated if an economic model has not taken spatial analyses into account. A spatial economic model can address this problem by introducing spatial factors [59]. Besides this, the variable of the current time may be affected by the variable of the previous period due to a change in the time inertia of the variable, thus the variable’s spatial dependence may be reflected in both time and space. There are three basic spatial econometric models—the spatial lag model (SLM), the spatial error model (SEM), and the spatial Durbin model (SDM). The SLM only includes the lag term of the spatial dependence variable, and the SEM only includes the spatial spillover effect of the independent variables, whereas the SDM includes both the lag term of the spatial dependence variable and the spatial spillover effects of the independent variables. Given that the SDM can be simplified to the SLM or the SEM under certain circumstances, this study incorporated the one-stage lag term of the explained variable into the dynamic spatial Durbin model (DSDM), in order to explore the influence of land dispositions on urban innovation. The specific DSDM can be expressed as follows [60]:(2)$${\mathrm{INNOV}}_{i,t}{=a+\varphi \;\mathrm{INNOV}}_{i,t-1}+\lambda \left({\mathrm{WINNOV}}_{i,t}\right)+\theta \left({\mathrm{WINNOV}}_{i,t-i}\right){+\rho }_{1}{\mathrm{TPD}}_{i,t}{+\rho }_{2}\left({\mathrm{WTPD}}_{i,t}\right)+\phi_{1}{\mathrm{PALA}}_{i,t}+\phi_{2}\left({\mathrm{WPALA}}_{i.t}\right){+\delta X}_{i,t}{+\mu }_{i}{+\nu }_{i}{+\varepsilon }_{i,t}\;$$
where INNOV*_i,t_* represents the urban innovation index increment of the *i*th prefecture-level city in the *t*th year; TPD*_i,t_* and PALA*_i,t_* are core explanatory variables and *X_i,t_* represents a group of control variables; *a* is the constant term; *φ* is the coefficient of the one-stage time lag term, representing how urban innovation in the *t−*1th year impacts urban innovation in the *t*th year; *λ* is the coefficient of the spatial lag term, reflecting the influence of neighboring cities’ innovation levels on local cities; *θ* is the coefficient of the time and spatial lag term, indicating the effect of a neighboring city’s innovation level in the *t*th year on a local city’s innovation in the *t−*1th year; *ρ*_1_ and *ϕ*_1_ are coefficients of explanatory variables; *ρ*_2_ and *ϕ*_2_ are coefficients of the spatial lag term of the explanatory variables, representing the impacts of a neighboring city’s land transfer price differences and its proportion of agreement-based land transfer area on a local city’s land dispositions; *μ_i_* and *ν*_i_ represent the spatial and time effects, respectively; *ε_i,t_* is the error term vector and *W* is the spatial weight matrix. In addition, when *φ* = 0, a static spatial panel model will be given; when *φ* = *λ* = *ρ*_2_ = *ϕ*_2_ = 0, it will be an ordinary panel model.

It is necessary to identify which model suits the data best by conducting a series of model tests [61]. First, this study employed the Lagrange multiplier (LM) test and robust Lagrange multiplier test, proposed by Anselin, to test whether the model we established contains spatial interactions [62]. Meanwhile, the likelihood ratio (LR) test should be conducted to test whether the spatial panel model is more suitable for this study, and the SDM can be simplified to the SLM or the SEM. At the same time, the Hausman test should be employed to select the fixed effect and the random effect of the model. If the Hausman test is significant, the fixed effect model is adopted. If not, the random effect model should be selected.

### 3.3. Data Source

A city is a primary center of innovation activities. This study focuses on the influence of land dispositions on the innovation of prefecture-level cities in mainland China. Located in the east of Asia and the west coast of the Pacific Ocean, China has a land area of about 9.6 million km^2^ and consists of 333 prefecture-level cities. And there may distinguished differences in geographical positions, natural resource endowment, economic conditions, industrialization, urbanization among different cities. To ensure the integrity, continuity, and availability of the data, cities with missing values and outliers were omitted. Therefore, the panel data of 266 prefecture-level cities from 2004 to 2017 were collected into a sample to conduct empirical research, and the locations of the study areas are showed in Figure 2.

The urban innovation index data were obtained from the “FIND Report on City and Industrial Innovation in China (2017)”. The GDP, general budgetary expenditure on R&D, number of ordinary higher colleges, actual foreign direct investment, and size of loan from the national banking system at the year-end were taken from the “China City Statistical Yearbook (2005–2018)”. The data on land transfer price and area as determined by “agreement, bidding, listing, leasing” were obtained from the “China Land and Resources Statistical Yearbook (2005–2018)”. Besides this, actual foreign direct investments were converted from USD into RMB using the average RMB exchange rate of the current year. Table 2 shows the descriptive statistics of 266 prefecture-level administrative units in China from 2004 to 2017, including the observation of sample, mean value, standard deviation, minimum value and maximum value.

## 4. Results

### 4.1. Results of Relevant Tests

#### 4.1.1. Results of the Spatial Autocorrelation Test

Since spatial correlation is one of the prerequisites for using a spatial economic model, the Global Moran’s I index was employed to verify whether spatial correlation exists in urban innovation among China’s prefecture-level cities. The variation trend in the Global Moran’s index of urban innovation can be seen in Figure 3. The Moran’s I indexes of urban innovation were greater than zero, and the corresponding P values were statistically significant and positive at the 10% level, except in 2004. This means that the urban innovation levels of China’s prefecture-level cities are spatially dependent, and obvious spatial agglomeration can be observed. Therefore, a spatial econometric model is required to investigate the problems studied in this paper.

#### 4.1.2. Results of Model Selection Test

Through the above analysis, it was determined that there is spatial correlation in the urban innovation level among prefecture-level cities. Therefore, we selected the most suitable model by performing the LM, robust LM, LR and Hausman tests. Table 3 reports the results of the LM test under space and time double-fixed effects, as well as the results of the LR test and the Hausman test. The results suggest that the LM test results regarding both the SLM and the SEM are significant at the 1% level. When using the robust LM test, a null hypothesis of no spatial lag effect was accepted at the 10% level, while a no spatial error effect was rejected, indicating that spatial dependence does exist in the data, and spatial panel models were more appropriate for the estimation. In the LR test, both null hypotheses: *θ* = 0 and *θ* + *ρβ* = 0 were significant at the 1% level, rejecting the assumption that SDM could be reduced to SLM and SEM, and demonstrating that the SDM is the most suitable for spatial panel data. Meanwhile, the results of the Hausman test indicate that the assumption that the model has random effects can be rejected at the 1% significance level, indicating that the SDM with spatial fixed effects should be adopted in this study [63].

### 4.2. Spatial Econometric Regression Results

Matlab 2014b and Stata 16 software were used to perform the regression analysis. Table 4 reports the estimated regression results of the static spatial Durbin model (SSDM) under time-fixed effects, the dynamic spatial Durbin model (DSDM) under time-fixed effects, the spatial-fixed effects, and the spatial and time-fixed effects. As we can see from Table 4, the estimate of the one-stage time lag of urban innovation (INNOV*_t_*_−1_) was significantly positive at the 1% level in both models (2) and (4). The spatial lag coefficients of urban innovation (W × INNOV) were found to be statistically significant across different models, except for model (3). The spatial lag term (W × INNOV*_t_*_−1_) of urban innovation was significantly positive in model (3), while it was significantly negative in model (4). Besides this, as a result of an integrated analysis of the estimated value of R-sq and Log-likelihood, as well as the economic logic implied by the estimated coefficient of the core explanatory variables, it is apparent that the DSDM under spatial and time-fixed effects had the optimal fitting specification in this study. Thus, the following analysis revolves around the estimated regression results of model (4).

The fifth column of Table 4 reports the regression results of the DSDM under spatially and time-fixed effects. In the time dimension, the estimate of the one-stage time lag of urban innovation (INNOV*_t_*_−1_) was significantly positive at the 1% level, suggesting that the development of innovation in prefecture-level cities in China is relevant over time. Urban innovation would continue to grow in the following year if the city performed well in its innovation development in the present year. Specifically, an average increase of 1% in urban innovation in the present year would drive a 1.307% increase in the next year. This time inertia reflects the strong growth momentum of urban innovation, which will occur with the rapid development of China’s industrialization and urbanization sectors. As for the spatial dimension, the spatial lag coefficient of urban innovation (W × INNOV) was also found to be statistically significant and positive at the 1% level; an average increase of 1% in innovation development in local cities would lead to a 0.413% increase in neighboring cities at the same time. This finding is consistent with the Moran’s I index in Figure 3, which confirms once again the existence of spatial dependence in urban innovation in China’s prefecture-level cites, as a 1% increase in urban innovation in one city is associated with a 0.406% increase in innovation in neighboring cities. In the time and space dimensions, the estimates of time and spatial lag in urban innovation (W × INNOV*_t_*_−1_) were statistically significant and negative, as an average increase of 1% in urban innovation in one city would lead to a 0.291% decrease in innovation in neighboring cities in the next year. This result indicates that the improvement of one city’s innovation level tends to inhibit the development of urban innovation in neighboring cities in the following year. In part, this is due to the Matthew effect that operates in regional innovation development. Innovation resources tend to be absorbed by cities showing better innovation performance, making it more and more difficult for neighboring cities, who are developing comparatively slowly, to effectively accumulate innovation resources.

As regards the two core explanatory variables, the estimates of the non-spatial and the spatial spillover of both the land transfer price departure (LTPD) and the proportion of agreement-based land transfer area (PALA) were all found to be significantly positive at the 1% level. More specifically, an average increase of 1% in LTPD in one city would lead to a 0.016% increase in innovation in that city, and a simultaneous 0.076% increase in neighboring cities. A 1% increase in PALA was associated with a 0.027% increase in innovation in the same city, and a 0.008% increase in innovation in neighboring cities. These results confirm that the two land disposition modes derived from land finance do have an impact on urban innovation in prefecture-level cities in China. In general, the promotional effects of LD mode1 and LD mode2 on urban innovation are stronger than the inhibitory effects. Such a conclusion shows that manufacturing enterprises still rely heavily on the financial support of the local government to remedy the negative effects of land dispositions. LD mode1 can help provide considerable land finance revenue for local governments, enabling them to offer stable funding for manufacturing enterprises, so that they can carry out innovation activities without concern for capital sources. LD mode2 is conducive to the improvement in infrastructure required for industrial production, thereby accelerating the development urban industrial foundations, as well as providing an innovation platform for manufacturing enterprises. In addition, the spatial spillover effects of urban innovation shown in LD mode1 and LD mode2 indicate that intense intergovernmental competition exists in almost every aspect of local developmental strategies. Seeking promotions, local government officials maintain a rivalry with their “peers” in promoting urban innovation and land development policies.

### 4.3. Spatial Effects Decomposition Analysis

As a result of the spatial spillover effect, the change in explanatory variable tends to affect both the explained variable of the one city and its neighboring cities, while in turn influencing the explained variable of the one city through the feedback effect. Therefore, the estimated coefficients of the regression models are not sufficiently rigorous and are unable to directly reflect the marginal effects of explanatory variables on the explained variable. To some extent, regression models can only provide valid results with regard to influence direction and statistical significance. With this in mind, the impacts of land dispositions on urban innovation can be further decomposed into direct and indirect effects. The direct effect is the impact exerted by explanatory variables on the innovation of a local city (*X*→*Y*) and the feedback effect is also embraced (*X*→*WY*→*Y*), while the indirect effect captures how the explanatory variables of one city affect urban innovation in neighboring cities, and this influence results in a spatial spillover effect in the explanatory variables (*X*→*WX*→*WY*). Furthermore, the direct and indirect effects can be further decomposed into short-term effects and long-term effects, since from when the DSDM is employed, the instant impact and the long-term impact generated by the time lag effect can be reflected, respectively. 

Therefore, we further decompose the total effects of the spatial model into direct effects and spatial spillover effects by means of the partial differentiation method. The results of this decomposition are summarized in Table 5. Overall, the absolute values of most long-term effect estimates were greater than the coefficients of the short-term effects, indicating that the impacts of most variables on urban innovation are profound and far-reaching. In terms of LTPD, the estimates of the direct, indirect, and total effects were found to be statistically significant and positive in the short term, while the coefficients were all significantly negative in the long term. The results suggest that LD mode1 has the same direction of influence on innovation in both local city and its neighboring cities, over both the short and the long term. More specifically, LD mode1 would have a strong positive impact on the development of urban innovation in a local city and its neighboring cities in the early stage, however, the promotional effects tend to give way to inhibitory effects over time. Such a finding agrees with the theoretical analysis in Section 2. Since investment and financing channels for innovative activities are bound to broaden in the long term, the dependence of manufacturing enterprises on financial support from the local government would decrease accordingly. In this way, the negative impacts of LD mode1 increase, with the most significant underlying problems being the crowding-out effect and innovation talent flight. Some of the funds provided for innovation projects would be diverted to real estate businesses, as many manufacturing enterprises succumb to the temptations of the huge profits derivable from real estate. The distribution of innovation funds would result in the crowding out of innovative activities, which would be inimical to the improvement of the urban innovation. Furthermore, innovation talent flight is a familiar phenomenon due to the high urban living costs generated by the over-developed property industry, which would cause significant damage to the human capital of high-technology companies.

Similarly, a positive impact of PALA on innovation development was also observed in both local and neighboring cities for the near future, while the long-term direct and total effects tend to be statistically significant and negative. One point that is worth noting is that the long-term spillover effect was found to have no statistical significance. The above results indicate that the effects of LD mode2 on urban innovation also change from promotional to inhibitory with time. On the one hand, LD mode2 is unquestionably beneficial to the establishment of a solid urban industrial foundation in the initial period of industrialization. This is due to the crowding-out and threshold effects triggered by LD mode2 filtering out traditional industries with low added value, whose place would be taken by relatively technology-intensive industries, thereby upgrading and restructuring the manufacturing industry. Besides this, many individuals with innovation talents gravitate towards more industrialized cities in order to enjoy the superior quality of life brought about by the industrial correlation effect, and the agglomeration of human capital promotes urban innovation. On the other hand, the industrial homogenization caused by LD mode2 would hinder industrial transformation and upgrades as the industrialization process is accelerated. Meanwhile, special treatment from local governments would reduce the motivation and activity pertaining to innovation in manufacturing enterprises. Overall, the inevitably adverse effects exerted by LD mode2 far overweigh the promoting effect. It is not surprising that local governments tend to adjust land disposition modes once the inhibitory effect on urban innovation becomes obvious and the spillover effect of the LD mode2 on neighboring cities begins to disappear over time.

As for the control variables, both the direct and indirect effects of economic development level (INPGDP) on urban innovation were found to be statistically significant and positive in the short term, while the positive effects would become negative over time. Our results suggest that the promotional effect of economic development on local and neighboring cities’ urban innovation would disappear, and eventually change into an inhibitory effect with time. The direct and indirect effects of the scale of investment in R&D (RDG), higher education level (EDU) and financial development level (FDL) on urban innovation show similar trends to the economic development level. One possible reason for this is that improvements in the local economy, the input into R&D, the higher education level, and the local financial situation might intensify the competition between neighboring governments, thereby giving rise to race to the bottom in innovation development in neighboring areas. In the long term, though, other pivot points of innovation development should be cultivated to enhance the innovation level in both local and neighboring cities. As for the degree of opening-up (OPE), the estimated direct effects were found to be statistically significant and negative in the short term, while this coefficient is insignificant in the long term, and the indirect effect shifts from significantly negative to positive with time. The results indicate that improvements in local industries as a result of improving openness occur slowly, but may lead to intergovernmental competition over investment invitations, and may therefore promote innovation development in neighboring cities in the long term.

### 4.4. Regional Empirical Analysis

Since coastal areas offer geographical advantages in terms of industrial development, the eastern region has taken the lead in development in the initial stage of industrialization. The impacts of land dispositions on urban innovation might have regional heterogeneity due to the uneven development of land finance and economic growth across the country. To gain a deeper understanding of the overall impact of land dispositions derived from land finance on urban innovation, analyses based on East and Midwest China have been conducted and their results compared. Table 6 and Table 7 provide comparisons of the regression results of the SSDM with spatially and temporally fixed effects, as well as the effect of decomposition results on two city groups; the results are very clearly quite different.

In terms of the time dimension, the time relevance of urban innovation was observed in both East and Midwest China, and the development trends in both areas show typical path dependence. However, the estimates suggest midwestern cities depend more on the selected path than eastern cities when promoting innovation. Specifically, an average increase of 1% in the innovation level in the present year would lead to a 1.069% increase in the next year in East China, and a 1.222% increase in Midwest China. Conversely, it was found that East China has a far greater spatial aggregating effect on urban innovation than Midwest China. The results show that a 1% increase in urban innovation was associated with a 0.697% increase in innovation in neighboring cities in the East, while the estimate of the spatial spillover effect among midwestern cities is just 0.130%. As for the effect of competition in urban innovation on the time–space perspective, the same inhibitory phenomena arose throughout eastern cities as were seen in the rest of the sample, while these were not observed in Midwest China. In the East, an average increase of 1% in urban innovation in a local city in the present year would lead to a 1.564% decrease in innovation in neighboring cities in the next year, however, the estimate of time and spatial lag in urban innovation in Midwest China was not statistically significant. In general, the development of urban innovation in these two areas has shown not only time inertia, but also spatial spillover effects. Since political positions in economically developed areas such as the east coast are more attractive, intergovernmental competition in the area of urban innovation was much fiercer among eastern cities than midwestern cities. The existence of the Matthew effect in regional innovation development was further verified in East China. Furthermore, eastern cities with good innovation performance are better at absorbing innovation resources, thus the inhibitory effect on the local city’s innovation level compared to neighboring cities in the previous year is stronger than anywhere else in the sample. One possible reason for the non-significant estimate of time and spatial lag in Midwest China is that, since the innovation capabilities of midwestern cities remain relatively low, most of them are still exploring how to motivate the initial steps of urban innovation, and the Matthew effect has not yet become apparent.

More importantly, many valuable insights have been derived from the comparative analysis of effect decomposition in the two areas. Both LD mode1 and LD mode2 effectively facilitate urban innovation in East and Midwest China in the short term, however, land dispositions resulting from land finance tend to contribute more to the development of innovation in eastern cities. The estimates of the direct and indirect effects of LTPD are 0.180 and 0.798 in East China, which are greater than those in Midwest China. This is also the case for the direct and indirect effects of PALA. The main cause of this phenomenon is that land finance first emerged and developed as the main source of financial revenue in the eastern coastal areas after the tax-sharing reform. The considerable land finance revenue generated by LD mode1 provided financial support for the construction of urban infrastructures, while the manufacturing capital attracted by LD mode2 was very helpful in creating opportunities for employment and economic growth. In turn, population agglomeration and the improvement of public infrastructure further increase the value of urban land, and create more competitive conditions for investment at the same time. As such, the two land disposition modes not only enabled the first level of capital accumulation needed to support rapid industrialization, but they also helped accumulate essential elements favorable for the promotion of urban innovation. Midwestern cities also followed this method of land resources utilization after the strategies for elevating central China and the large-scale development of western China were launched. In order to raise the value of urban land resources and attract industrial business and talent capital as quickly as possible, the construction of new urban districts was vigorously promoted in the Midwest. Though such land disposition modes increased the local revenues of midwestern cities to a certain extent, and greatly improved the urban infrastructure and investment invitation conditions, they failed to achieve the same facilitative effect on urban innovation as was seen in eastern cities. For one thing, despite LD mode1 having led to massive collective land changes in the countryside in the name of national construction via land urbanization, the lack of housing demand has led to a mismatch between population urbanization and land expansion. Meanwhile, the investment policies of midwestern cities did not live up to expectations, even though an accommodative environment was established. The strong competitive industrial clusters in eastern areas hinder industries “westward”, and the regional difference is thus further enlarged.

In the long term, the facilitative effect of eastern cities’ land dispositions on urban innovation are tending to decrease, while both LD mode1 and LD mode2 appear to inhibit the development of urban innovation in Midwest China. Specifically, the estimates of the direct and indirect effects of LTPD and PALA were statistically non-significant, and the total effect continues to decrease with time in eastern cities. Meanwhile, the estimates of direct and indirect effects were all found to be statistically significant and negative in mid-western areas. Competition over traditional manufacturing industries between eastern and midwestern cities is a major factor that cannot be ignored here. At present, traditional manufacturing industries remain a central pillar of the economy of East China, since they are still not easily replaced by innovation-intensive industries on a large scale. Thus, rivalry within mid-low-end industries between eastern and mid-western cities is almost inevitable. Due to their superior natural geographic advantages, as well as the talent and capital advantages they accumulated in early stages, capital investments in the manufacturing industry tend to gather in eastern cities, and the labor force of the Midwest is also being drawn to the East. In addition, the facilitative effects of the two modes on eastern cities’ innovation have diminished with time; this could be due to the fact that more economically developed areas had already accumulated a certain amount of the capital required for initial innovation development, and an urban industrial foundation has been established and improved in these areas. As such, the contribution of the land disposition mode is now limited. If eastern cities continue to adopt LD mode1 and LD mode2 to encourage further growth in their economies, their development of a regional industrial structure founded on innovation will be restricted. Furthermore, they might be forced to compete for the resources required for low-end industries with midwestern cities, which would aggravate the depopulation problem in the Midwest. In this way, mid-western cities will come up against difficulties in the restructuring and upgrading of their industrial foundations, and their urban development will be inhibited due to the lack of various innovation resources.

## 5. Conclusions and Recommendations

The implementation of the tax-sharing reform and the full opening of the land transfer market have given rise to land finance, and two land disposition modes have gradually taken shape in China. This study depicts in detail the paths of the positive and negative effects of LD mode1 and mode2 on urban innovation, and then further investigates the theoretical mechanisms by employing a DSDM based on panel data covering 266 prefecture cities from 2004 to 2017.

The following key findings emerged: (1) The development of innovation was path-dependent, and the spatial agglomeration of urban innovation was confirmed. Meanwhile, the innovation performance of one city was found to inhibit the innovation levels of neighboring cities in the following year due to the Matthew effect. These phenomena were also evident in eastern areas, while the spillover effect was not obvious in mid-western cities; (2) In general, the combined impacts of land dispositions on urban innovation underwent a change from facilitative in the early stage to inhibitory at the current stage. The influence direction of LD mode1 on the urban innovation of local and neighboring cities was consistent, showing that positive impacts will be offset by negative impacts over time. The influence of LD mode2 followed a similar dynamic path, although here, the spillover effect on neighboring cities was shown to disappear in the long run; (3) Since the independent innovation capacities of eastern cities have already improved in the past few years, they depend less on government support to begin their innovation activities. Thus, the two LD modes could only help a little with urban innovation, and the facilitative effect tends to disappear gradually; (4) The impact of land disposition on urban innovation in mid-western cities will change from promotional to inhibitory with time. Although the two LD modes can help underdeveloped cities accumulate innovation funds and form industrial foundations in the early stage, the facilitative effects would be offset by the fact that these areas are unappealing to innovation resources.

The following policy implications can be drawn from the above findings: (1) The land transfer marketization reform should be moved forward to break the local government’s monopoly on land supply. The supply of different types of urban land is supposed to dynamically adjust itself, according to the real demands for urban development; (2) A rational structure for the supply of urban land should be provided to amend land disposition modes derived from land finance. On the one hand, industrial land supply should be curtailed so that urban land use efficiency and the settlement threshold of traditional manufacturing enterprises can be improved. That said, residential and commercial land supply should be expanded to control the high land costs and soaring real estate prices, which will not only drive down urban living costs, but will also reduce the dependence of the local government on land finance. Meanwhile, a property tax ought to be introduced as a stable tax source; (3) The universal solution should be replaced with distinct land disposition strategies according to the actual requirements of different regions, and the specific conditions and strategies for the innovation-driven development of different cities should be taken into consideration; (4) The performance evaluation system for local officials should be reformed. A GDP-oriented promotion system is not conducive to the improvement of cities with lower innovation capacities. Indexes related to urban innovation should be included, and economic indicators should be made less prominent to limit excessive inter-governmental competition.

This study has several limitations, which could be remedied in further research. The evaluation indicators of the two LD modes should be diversified; it would be helpful to carry out further in-depth analyses based on a comprehensive database, via which more diverse types of land transaction data could be included. Besides this, this study only reflects the effects of land disposition on urban innovation during relatively early stages, due to the unavailability of some indicators. It might make more sense to conduct further empirical analyses by incorporating more recent data, thus yielding a result that fits better.

## Figures and Tables

**Figure 1 ijerph-19-03212-f001:**
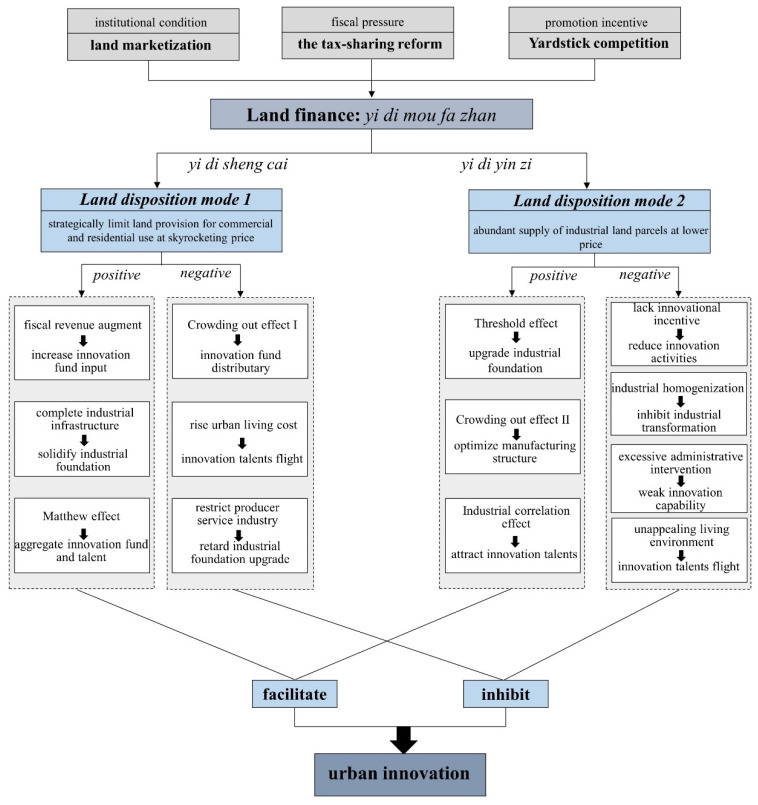
Paths of influence of land dispositions derived from land finance on urban innovation.

**Figure 2 ijerph-19-03212-f002:**
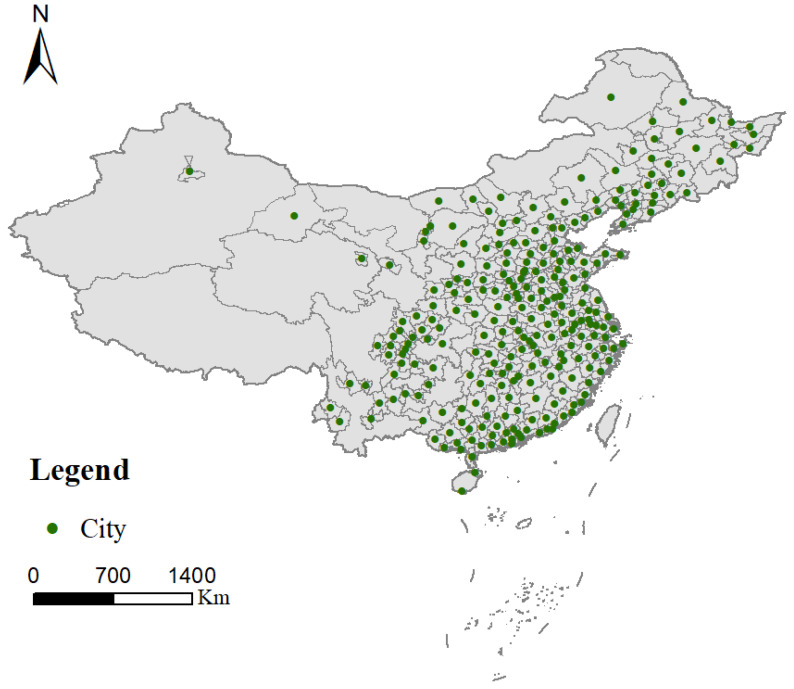
The locations of 266 prefecture-level cities in China.

**Figure 3 ijerph-19-03212-f003:**
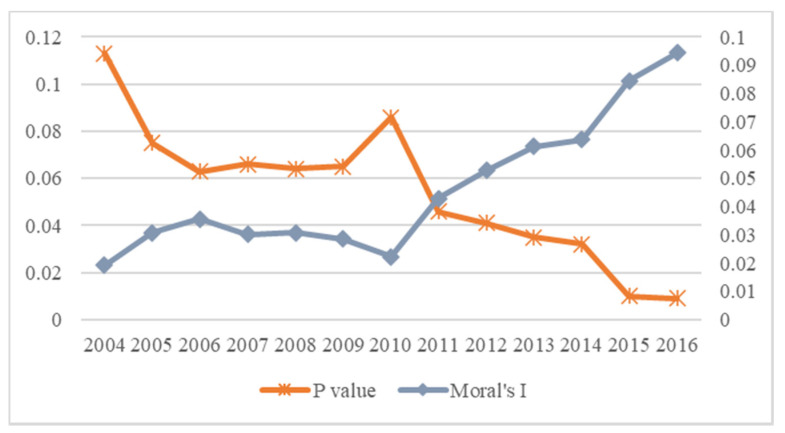
Moran’s I index of urban innovation from 2004 to 2016.

**Table 1 ijerph-19-03212-t001:** Variable selection and explanation.

Variable	Expression	Explanation
Dependent Variable		
Urban innovation index	INNOV	Urban innovation index increment. The single patent value is evaluated according to patent duration, then by adding up all patent values for each city in order to obtain the cumulative value; one subtracts the indexes for the current and previous years to get the urban innovation index increment.
Independent Variable		
Land transfer price departure	LTPD	The proportion of market-oriented land transfer price with agreement-based land transfer price.
Proportion of agreement-based land transfer area	PALA	The proportion of agreement-based land transfer area to the total land transfer area.
Control Variable		
Economic development level	INPGDP	The logarithm of per capita GDP.
Investment scale of R&D	RDG	The proportion of R&D investment in GDP.
Higher education level	EDU	The number of ordinary colleges in each city.
Degree of the opening-up	OPE	The proportion of actual direct foreign investment in GDP.
Financial developing level	FDL	The proportion of the loan size from the national banking system at year-end in GDP.

**Table 2 ijerph-19-03212-t002:** Descriptive statistics of the variables.

Variable	Obs.	Mean	Std. Dev.	Min	Max
INNOV	3458	2.1950	10.593	−0.2880	289.0720
LTPD	3458	4.3232	9.7654	0.0034	285.8316
PALA	3458	0.2451	0.2672	0.000014	0.9858
INPGDP	3458	10.2341	0.7848	7.8474	15.6752
RDG	3458	0.0035	0.0059	0.0000028	0.0660
EDU	3458	8.0000	14.0000	0.0000	92.0000
OPE	3458	0.0027	0.0279	0.000002	0.7752
FDL	3458	1.0673	0.8355	0.1122	16.7426

**Table 3 ijerph-19-03212-t003:** Results of spatial econometric model tests.

Determinants	Statistics	Determinants	Statistics
LM test spatial lag	37.819 *** (0.00)	Robust LM test spatial error	0.795 (0.37)
Robust LM test spatial lag	2.956 * (0.09)	LR test spatial lag	2388.804 *** (0.00)
LM test spatial error	35.657 *** (0.00)	LR test spatial error	262.624 *** (0.00)
Hausman	59.487 *** (0.00)		

Note: The *p*-values for the coefficients in parentheses, ***, ** and *, are significant at the 1%, 5%, and 10% levels, respectively.

**Table 4 ijerph-19-03212-t004:** Estimated regression results of the SDM.

Determinants	(1) SSDM (Time-Fixed Effects)	(2) DSDM (Time-Fixed Effects)	(3) DSDM (Spatial-Fixed Effects)	(4) DSDM (Spatial- and Time-Fixed Effects)
INNOV*_t_*_−1_	—	1.113 *** (310.67)	—	1.307 *** (260.16)
W × INNOV	−0.236 *** (−10.30)	0.423 *** (75.26)	0.012 (0.36)	0.413 *** (17.46)
W × INNOV*_t_*_−1_	—	—	0.445 *** (7.73)	−0.291 *** (−9.37)
LTPD	0.026 (1.64)	−0.029 *** (−11.27)	0.001 (0.07)	0.016 *** (5.88)
W × LTPD	0.004 (0.11)	−0.059 *** (−10.90)	0.009 (0.36)	0.076 *** (13.32)
PALA	0.064 *** (6.86)	0.0219 *** (13.17)	0.013 (1.70)	0.027 *** (14.45)
W × PALA	0.006 *** (0.51)	0.072 *** (25.88)	−0.013 (−1.30)	0.008 *** (2.59)
INPGDP	1.959 *** (6.57)	0.265 *** (4.08)	−3.219 *** (−4.63)	16.970 *** (102.91)
W × INPGDP	1.720 *** (3.27)	6.538 *** (74.33)	2.587 ** (3.32)	71.77 *** (289.17)
RDG	6.334 *** (14.52)	9.390 *** (78.39)	4.550 *** (7.43)	9.713 *** (68.33)
W × RDG	0.220 *** (0.31)	41.600 *** (227.46)	0.893 (0.98)	52.330 *** (241.22)
EDU	0.340 *** (26.70)	0.022 *** (9.13)	0.694 *** (13.25)	0.400 *** (32.22)
W × EDU	0.123 *** (4.49)	0.007 *** (1.38)	−0.041 (−0.45)	1.124 *** (53.36)
OPE	−0.165 *** (−2.81)	−0.257 *** (−23.35)	−0.054 (−0.93)	−0.049 *** (−3.68)
W × OPE	−0.208 ** (1.96)	−1.582 *** (−84.06)	−0.479 *** (−4.71)	−0.973 *** (−40.44)
FDL	−0.014 *** (−5.72)	−0.0125 *** (−27.15)	−0.023 *** (−9.45)	0.020 *** (33.94)
W × FDL	−0.001 (−2.56)	−0.029 *** (−41.42)	0.024 *** (7.39)	0.060 *** (58.35)
R-sq	0.282	0.308	0.351	0.362
N	3458	3458	3458	3458

Note: The *t*-values for the coefficients in parentheses, ***, ** and *, are significant at the 1%, 5%, and 10% levels, respectively.

**Table 5 ijerph-19-03212-t005:** Coefficients of spatial effects’ decomposition of the SSDM.

Determinants	Short Time			Long Time		
	Direct Effects	Indirect Effects	Total Effects	Direct Effects	Indirect Effects	Total Effects
LTPD	0.026 *** (9.18)	0.132 *** (12.62)	0.158 *** (13.22)	−0.031 ** (−3.16)	−0.184 *** (−10.59)	−0.215 *** (−11.74)
PALA	0.003 *** (15.68)	0.030 *** (5.78)	0.059 *** (9.93)	−0.088 *** (−13.67)	0.007 (0.72)	−0.080 * (−9.50)
INPGDP	26.470 *** (34.28)	125.500 *** (23.52)	152.000 *** (24.95)	−35.28 *** (−9.39)	−171.2 *** (−23.14)	−206.400 *** (−18.67)
RDG	16.501 *** (29.78)	89.700 *** (24.28)	106.200 *** (25.09)	−16.740 *** (−5.87)	−127.5 *** (−25.47)	−144.300 *** (−18.56)
EDU	0.555 *** (30.33)	2.056 *** (21.25)	2.611 *** (23.52)	−1.005 *** (−14.67)	−2.542 *** (−16.75)	−3.547 *** (−17.51)
OPE	−0.169 *** (−11.11)	−1.582 *** (−22.95)	−1.751 *** (−22.53)	−0.134 (−1.77)	2.513 *** (25.65)	2.379 *** (17.06)
FDL	0.028 *** (31.24)	0.109 *** (21.09)	0.137 *** (23.31)	−0.049 *** (−14.00)	−0.137 *** (−17.39)	−0.186 *** (−18.29)

Note: The *t*-values for the coefficients in parentheses, ***, ** and *, are significant at the 1%, 5%, and 10% levels, respectively.

**Table 6 ijerph-19-03212-t006:** Results of regression and effects decomposition for East China.

Determinants	DSDM Result	Short Term			Long Term		
		Direct Effects	Indirect Effects	Total Effects	Direct Effects	Indirect Effects	Total Effects
INNOV*_t_*_−1_	1.069 *** (136.77)						
W × INNOV	0.697 *** (19.57)						
W × INNOV*_t_*_−1_	−1.564 *** (−33.40)						
LTPD	0.098 *** (19.55)	0.180 *** (14.30)	0.798 *** (7.28)	0.978 *** (8.06)	2.213 (0.06)	−1.848 (−0.05)	0.365 *** (17.19)
W × LTPD	0.191 *** (17.52)						
PALA	0.099 *** (23.74)	0.135 *** (18.46)	0.352 *** (6.13)	0.487 *** (7.64)	1.805 (0.06)	−1.624 (−0.05)	0.181 *** (15.07)
W × PALA	0.046 *** (6.56)						
INPGDP	14.250 *** (29.16)	77.890 *** (9.92)	618.300 *** (8.13)	696.200 *** (8.30)	726.400 (0.08)	−466.800 (−0.05)	259.600 *** (22.54)
RDG	26.73 *** (71.44)	89.160 *** (11.39)	606.300 *** (8.02)	695.400 *** (8.33)	904.000 (0.07)	−644.700 (−0.05)	259.400 *** (22.78)
EDU	2.025 *** (61.74)	4.575 *** (13.84)	24.770 *** (7.75)	29.350 *** (8.32)	51.300 (0.07)	−40.360 (−0.05)	10.940 *** (22.52)
OPE	0.050 (1.22)	−0.820 *** (−7.28)	−8.442 *** (−8.24)	−9.262 *** (−8.20)	−6.986 (−0.11)	3.532 (0.05)	−3.454 *** (−20.60)
FDL	0.063 *** (40.16)	0.104 *** (17.54)	0.395 *** (7.14)	0.499 *** (8.18)	1.331 (0.06)	−1.145 (−0.05)	0.186 *** (20.63)
R-sq	0.327						
N	1469						

Note: The *t*-values for the coefficients in parentheses, ***, ** and *, are significant at the 1%, 5%, and 10% levels, respectively.

**Table 7 ijerph-19-03212-t007:** Results of regression and effects decomposition for Midwest China.

Determinants	DSDM Result	Short Term			Long Term		
		Direct Effects	Indirect Effects	Total Effects	Direct Effects	Indirect Effects	Total Effects
INNOV*_t_*_−1_	1.212 *** (151.76)						
W × INNOV	0.130 *** (4.27)						
W × INNOV*_t_*_−1_	0.062 (1.54)						
LTPD	0.002 (1.47)	0.002 (1.45)	0.006 * (2.47)	0.008 ** (3.20)	−0.015 * (−2.12)	−0.058 * (−2.32)	−0.073 * (−2.55)
W × LTPD	0.007 ** (2.58)						
PALA	0.004 ** (4.74)	0.004 *** (4.70)	0.002 (1.61)	0.006 *** (4.49)	−0.020 *** (−4.92)	−0.029 * (−2.05)	−0.050 ** (−2.99)
W × PALA	0.003 * (2.04)						
INPGDP	2.038 *** (32.42)	1.802 *** (23.12)	6.159 *** (32.03)	7.961 *** (34.29)	−13.93 *** (−5.90)	−55.510 *** (−3.79)	−69.440 *** (−4.09)
RDG	0.728 *** (12.66)	0.616 *** (9.79)	3.003 *** (29.33)	3.619 *** (30.85)	−5.475 *** (−4.92)	−26.100 *** (−3.91)	−31.570 *** (−4.07)
EDU	0.051 *** (8.98)	0.047 *** (8.24)	0.085 *** (10.53)	0.132 *** (14.06)	−0.305 *** (−6.32)	−0.851 ** (−3.29)	−1.156 *** (−3.85)
OPE	0.022 *** (4.45)	0.020 *** (3.94)	0.070 *** (7.58)	0.090 *** (9.82)	−0.153 *** (−4.50)	−0.627 ** (−3.59)	−0.780 *** (−3.89)
FDL	0.002 *** (8.24)	0.002 *** (7.34)	0.005 *** (11.98)	0.007 *** (15.22)	−0.013 *** (−5.60)	−0.047 *** (−3.67)	−0.060 *** (−4.02)
R-sq	0.365						
N	1989						

Note: The *t*-values for the coefficients in parentheses, ***, ** and *, are significant at the 1%, 5%, and 10% levels, respectively.

## Data Availability

Data available on request. The detailed experimental data presented in this study are available on request from the corresponding author.

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
