# Peer review of "The Influence of Land Disposition Derived from Land Finance on Urban Innovation in China: Mechanism Discussion and Empirical Evidence"

_ijerph, 2022, doi:10.3390/ijerph19063212_

Round 1

Reviewer 1 Report

The influence of land disposition derived from land finance on urban innovation in China: Mechanism discussion and empiri-cal evidence

Line 49. Describe between a parenthesis the meaning of GDP and RMB after the acronym (specifically RMB). Also, check the numbers that you put after “RMB”; I suppose that they are a number on “scientific notation”.

Line 50. I suggest to put a “;” instead of a coma after “… 28.4 times”. I leave it to your consideration.

Line 51. Remove “Significantly”, not necessary at this line.

Line 163. Remove “And”.

Line 174. Instead of a “.” after “…promotion chance”, employee a coma to continue with the “and”.

Line 353. Particularly, in this part you are repeating information (in some parts) in the text that is explained in the Table 1, and the readers of your work might be confused. Avoid that by doing a general description in the text (not repetitive, but summarized), and let the Table just as you have it.

Line 410. Before this line, you employee the “R&D” acronym. What does it mean? Clarify this in the previous lines (where you start using it).

Line 423. Add some information about the Moran´s I index. I don´t see any cite about why to use it.

Line 464. Hausman test is only mentioned in the Results part. It should be mentioned in the Materials and methods section.

Author Response

    We are grateful for the time you have invested in the thoroughly assessment of our manuscript and for providing such valuable comments. Those comments are all valuable and very helpful for revising and improving our paper, as well as the important guiding significance to our researches. Moreover, we have studies comments carefully and have finished the corrections. We hope it meets with approval. The main corrections in the paper and the responds to your comments are as follows.

(We reply to the question highlighted with red color, the added context and significant change in revised manuscript highlighted with yellow in revised manuscript.)

Point 1: Line 49. Describe between a parenthesis the meaning of GDP and RMB after the acronym (specifically RMB). Also, check the numbers that you put after “RMB”; I suppose that they are a number on “scientific notation”.

Response 1: We are thankful to the respected reviewer for pointing out this problem. As advised, the meaning of GDP and RMB have been described between a parenthesis after the acronym.

  • GDP(Gross Domestic Product). Page 2, line 49.
  • RMB(Renminbi). Page 2, line 50.

And thank you for reminding us of the formal error of numbers after “RMB”, they truly are a number on “scientific notation”. Their correct form have been remedied in the revised version of the manuscript. Under the guidance of your comment, we have thoroughly checked the whole manuscript to avoid making the same mistake.

  • 35×1013. Page 2, line 50.
  • 16×1013. Page 2, line 50.

Point 2: Line 50. I suggest to put a “;” instead of a coma after “… 28.4 times”. I leave it to your consideration.

Response 2: We are thankful to the respected reviewer for the valuable suggestion. As advised, we have changed the coma after “… 28.4-fold”to a semicolon in Page 2, line 50.

Point 3: Line 51. Remove “Significantly”, not necessary at this line.

Response 3: We are thankful to the respected reviewer for pointing out this problem. As advised, we have removed “Significantly” in Page 2, line 51.

Point 4: Line 163. Remove “And”.

Response 4: We are thankful to the respected reviewer for pointing out this problem. As advised, we have removed “And” in Page 4, line 163.

Point 5: Line 174. Instead of a “.” after “…promotion chance”, employee a coma to continue with the “and”.

Response 5: We are thankful to the respected reviewer for pointing out this problem. As advised, we have employed a coma instead of a full stop to continue with the “and” in Page 4, line 168.

Point 6: Line 353. Particularly, in this part you are repeating information (in some parts) in the text that is explained in the Table 1, and the readers of your work might be confused. Avoid that by doing a general description in the text (not repetitive, but summarized), and let the Table just as you have it.

Response 6: We are thankful to the respected reviewer for the carefully attention. We apologize for the confusion caused. Considering this excellent comment, we have removed the repeating information and made a general descripition in the revised manuscript.

As the dependent variable, the urban innovation index from the FIND Report on City and Industrial Innovation in China (2017) was selected to quantify the overall level of urban innovation [50]. We also calculated the value of urban innovation in the fol-lowing year using the growth rate to enlarge the sample. Based on the patent data and the micro–big data of the China Intellectual Property Office, and the registered capital data of the China Industrial and Commercial Bureau, the urban innovation index was calculated using the patent update model developed by Kou and Liu. It is worth men-tioning that although many studies employed the number of patents applied or au-thorized to measure the innovation level of a city [43,51-54], there are limitations to this approach. This is because statistical errors are inevitable when searching patent infor-mation manually, and it is hard to infer the true value of urban innovation by only considering the number of patents. Page 8, line358-367.

For the control variable, a group of indicators related to urban innovation were screened. 1) Economic development level. The marketization level increases with the development of the urban economy, and an active market economy would more effec-tively drive innovation investment. 2) Investment scale of Research and Development (R&D). This can determine the level of support via innovation funding given by local governments, and strongly influences the development of urban innovation. 3) Higher education level. The development of higher education leads to the expansion of high-quality talent capital, and industry–university–research institution cooperation has a beneficial stimulating effect on innovation activities. 4) Degree of the opening-up. An increase in opening-up is conductive to promoting industrial development and further improving the urban innovation level [2]. 5) Financial development level. In-novation projects often face financial restrictions due to the nature of venture activities, while financial support from national banking systems can to some extent relieve in-novation funding pressures [19]. Page 8-9, line380-392.

Point 7: Line 410. Before this line, you employee the “R&D” acronym. What does it mean? Clarify this in the previous lines (where you start using it).

Response 7: We are thankful to the respected reviewer for pointing out this problem. “R&D” is the acronym of “Research and Development”, and we have used its full form on their first appearance in the revised manuscript.

Investment scale of Research and Development(R&D). Page 8, line 382-383.

Point 8: Line 423. Add some information about the Moran´s I index. I don´t see any cite about why to use it.

Response 8: We are thankful to the respected reviewer for pointing out this problem. As advised, we have further enriched the basic content of the Moran´s I index and explained the reason it used in the revised manuscript. And we also added references cited in this part.

3.2.1. Global Moran’s Index

The innovation development of one city may show similar variation tendencies to neighboring cities due to the spillover and diffusion effects; such correlation between neighboring spatial units is called the spatial correlation [56]. Constructing a spatial econometric model is the main approach to investigating spatial correlation in some of the previous studies [8,13,30,57]. However, it is first necessary to examine whether the dependent variable has spatial dependence before conducting spatial economic analy-sis. The Global Moran’s I index is the most widely used method to measure whether a spatial correlation exists, and this reflects the degree of similarity between the attribute values of spatially adjacent or neighboring areas. The value is range [-1,1]. Greater than zero means the given variable shows positive spatial connections among its observa-tions and the existence of cluster in the spatial arrangement. Smaller than zero indicates that the given variable has negative spatial connection. When the value is zero, there is no spatial connection among the observation variables, and a random distribution in space can be inferred [58,59]. The calculation formula is as follows:

                      (1)

In Equation (1),  and  denote the urban innovation level of  city and  city respectively;  is the number of cities;  is the spatial weight matrix;  is the mean value of the samples; and  is the variance of the samples. In this study, a binary contiguity spatial weight matrix was constructed to reflect spatial adjacency. When region  is adjacent to region , =1; when region  is not adjacent to region , =0. Page 9-10, line 395-415.

Cited reference:

56.   Odland, J. Spatial Autocorrelation. Sage Publication, Newbury Park, CA, USA, 1988. Page 23, line 938.

8.     Fan, J.; Zhou, L. Three-dimensional intergovernmental competition, and urban sprawl: Evidence from Chinese prefec-tural-level cities. Land Use Policy 2019, 87, 104035. Page 22, line 848-849.

13.   Wang, J.; Wu, Q.; Yan, S.; Guo, G.; Peng, S. China’s local governments breaking the land use planning quota: A strategic interaction perspective. Land Use Policy 2020, 92, 104434. Page 22, line 858-859.

30.   Liu, Y. Government extraction and firm size: local officials’ responses to fiscal distress in China. J. Comp. Econ. 2018, 46, 1310-1331. Page 22, line 891-892.

57.   Lu, X.; Wang, M.; Tang, Y. The spatial changes of transportation infrastructure and its threshold effects on urban land Use Efficiency: Evidence from China. Land. 2021, 10, 346. Page 23, line 939-940.

Point 9: Hausman test is only mentioned in the Results part. It should be mentioned in the Materials and methods section.

Response 9: We are thankful to the respected reviewer for pointing out this problem. As advised, the reason why Hausman test and other tests should be conducted have been added in the Materials and methods section. And the results of Hausman test and other tests were presented in the Results part. The revisions are as follows.

It is necessary to identify which model suits the data best by conducting a series of model tests [62]. First, this study employed the Lagrange multiplier (LM) test and robust Lagrange multiplier test proposed by Anselin to test whether the model we established contains spatial interactions [63]. Meanwhile, the likelihood ratio (LR) test should be conducted to test whether the spatial panel model is more suitable for this study, and the SDM can be simplified to the SLM or the SEM. At the same time, the Hausman test should be employed to select the fixed effect and the random effect of the model. If the Hausman test is significant, the fixed effect model is adopted. If not, the random effect model should be selected.Page 10, line 449-457.

We sincerely appreciate the reviewers’ valuable comments on this article!

Reviewer 2 Report

The manuscript on "The influence of land disposition derived from land finance on urban innovation in China: Mechanism discussion and empirical evidence " by Han et al. presents an interesting study describing the impacts of land disposition derived from land finance on urban innovation, incorporating the government-governing mechanism in China. The presented data is interesting, and of high publication potential, but some minor improvements should be done by authors. My main concerns related to:
- the language quality (many sentences are definately too long, thus hard to follow), so I recommend revision by Native Speaker. 
- The Introduction chapter is too long, at leats one paragraph should be removed. 
- please higlight the novelty of presented study, it is totally omitted it in the current form. 
- The quality of the figure 1 should be imporved, it is hard to read the  text in the boxes.

After the revision of the text, please double check that all references are cited within the text, and that all citations within the text have a corresponding reference. Double check the spelling of the author names. Please unify the format of the reference list according to the Guide for Authors.

Author Response

We are grateful for the time you have invested in the thoroughly assessment of our manuscript and for providing such valuable comments. Those comments are all valuable and very helpful for revising and improving our paper, as well as the important guiding significance to our researches. Moreover, we have studies comments carefully and have finished the corrections. We hope it meets with approval. The main corrections in the paper and the responds to your comments are as follows. (We reply to the question highlighted with red color, the added context and significant change in revised manuscript highlighted with yellow in revised manuscript.)

Point 1: the language quality (many sentences are definately too long, thus hard to follow), so I recommend revision by Native Speaker.

Response 1: We are thankful to the respected reviewer for pointing out this problem. As advised, we have improved the English by a professional native proofreading from MDPI.

Point 2: The Introduction chapter is too long, at leats one paragraph should be removed.

Response 2: We are thankful to the respected reviewer for pointing out this problem. As advised, we have reorganized the Introduction chapter and simplified the literature review part frome three paragraph to two paragraph. The whole chapter has been reduced by about 20% of the original version.

Land finance is generally employed by local governments to augment fiscal reve-nue in China, which is characterized by the heavy reliance of local governments on land-related income (such as land transfer, lease, and tax fees) [1-3]. Since the tax-sharing reform in 1994, it has become urgent for local governments to seek ex-tra-budgetary revenue in order to relieve financial pressure as well as promote local development [4,5]. With land finance as an arguably irreplaceable driver, land-centered urbanization and industrialization (yi di mou fa zhan) have both contributed notably to China’s remarkable economic performance in the past few decades [6,7]. Ensuring an abundant supply of industrial land parcels at lower price in order to attract investments (yi di yin zi), and strategically limiting land provision for commercial and residential use at very high prices in order to collect funds (yi di sheng cai), are two prevalent land disposition modes derived from land finance, and they in turn greatly help local gov-ernments to maximize their land finance [2,3,8-14].

Underpinned by land finance, the previous round of China’s economic develop-ment has been distinguished by high-speed growth, and is widely considered “a growth miracle” [15,16]. The GDP (Gross Domestic Product) of the whole country had soared from 0.35×1013 RMB (Renminbi) in 1993 to 10.16×1013 RMB in 2020, an increase of more than 28.4-fold; China’s share of world GDP was only 1.72% in 1993, and it hit a historic high of about 17% in 2020. However, this high-speed economic growth, driven by nat-ural resources, is unsustainable in the long run due to the limited nature of land re-sources. Only by transforming such extensive development patterns can the country maintain this momentum in economic growth, and successfully enter the stage of high-quality development [2,17]. Accordingly, the central government developed the Outlines on National Strategy for Innovation-Driven Development in 2016, aiming at developing a new economic growth mode driven by knowledge and technology to preserve competitiveness and realize sustainable development [18]. The report of the 19th congress of the Communist Party of China highlighted that innovation is the pri-mary engine of development, and it is essential to develop new sources of economic growth driven by innovation so that the quality and efficiency of economic development can continue to steadily increase.

It is worth noting that, although the central government has devoted much energy to promoting innovation and inspiring participation in innovation activities, the tran-sition of the economic growth pattern from land-centered to innovation-driven remains difficult [19]. As regards the root cause, the path dependence of the local development mode and land finance are inseparably intertwined. More specifically, intergovern-mental competition has been encouraged by the central government to stimulate eco-nomic development by constructing fiscal and political incentives [20]. Even though this governance mechanism has made progress in promoting economic growth, it has also increased the dependence of local governments on land finance, and led to local officials’ blind pursuit of their short-term economic interests and vanity projects. For the sake of risk aversion, innovative projects characterized by high risks, uncertainty, and longer take-off periods would usually not be prioritized in local development strategies [16,19,21]. In part, the lack of support from local governments is a major reason why innovation-driven development is relatively slow in China [2]. In this pivotal stage of economic transformation and upgrading, it is of great practical importance to clarify what effects the land-centered development pattern might have on China’s urban in-novation development.

The existing studies that have shed light on this area can be categorized into the following two areas. The first study the relationship between land finance and land disposition in China. In the context of a GDP-based promotion appraisal system, the land transfer behavior of local governments is aimed at obtaining as much land finance revenue as possible in order to develop the local economy. Thus, two land disposition modes have developed from this heavy dependence on land finance: supplying limited commercial and residential land at higher price by means of bidding, auctions and listings, so as to amass huge capital [4,8,10,22], and leasing out large swathes of indus-trial land at lower prices through negotiation, so as to attract to investments with long-term tax benefits [3,8,10,11,13,23]. However, the tendency to thus seek uneven economic growth that neglects long-term social benefits should not to be overlook [8,14]. Secondly, the number of studies that have discussed how land disposition affects the relevant elements of innovation development has increased in the past few years. One of the most significant effects of this is that local governments can strongly support innovation activities by providing fiscal subsidies, as well as improving infrastructure. However, this is impossible without the high returns generated by commercial and residential land transfers [9,20,24]. Another important positive impact is the agglom-eration effect of manufacturing enterprises. In view of the mobility of industrial busi-ness, local governments should undertake a low-price transfer strategy with industrial land so as to attract enterprises into urban development zones and industrial parks. The effect of this would be that the industrial structure of a whole city can be upgraded by taking advantage of the technology transfer and knowledge spillover resulting from industrial agglomeration [2,14,25].

Only a small number of scholars have probed the direct influence of land disposi-tion, as derived from land finance, on overall innovation development. Yan et al. found that there is a significant inverted U-curve relationship between land finance and re-gional innovation at the provincial level—land finance promotes regional innovation at the beginning, but jeopardizes it in the end [26]. For prefecture-level cities, Xie discov-ered that land resource misallocation significantly reduces a city’s innovation capacity, and a higher ratio of land occupied by the supply areas of industrial and mining ware-houses leads to a lower innovation capacity [19]. Similarly, Xie and Hu verified the comprehensive negative effect of the current land resource allocation approach on ur-ban innovation in China [24]. While the above studies provided some empirical evidence to further explore the mechanisms by which land disposition influences urban innova-tion, there are three major limitations. Firstly, although these studies have roughly de-picted the way land finance, and the resulting land disposition, affect urban innovation, there is still a limited understanding of all the manifestations and details of the possible influence paths. Second, most of the analyses were carried out from a static perspective, and attempts to track the impacts at different stages of economic growth in China are lacking. Third, the uneven economic development in different regions of China has not been considered, and the regional differentiations in the causes of the impacts of land disposition on urban innovation ought to be taken seriously.

This study aims to deeply investigate the impacts of land disposition as derived from land finance on urban innovation, by incorporating government-governing mechanism in China. The data cover 266 Chinese prefecture-level cities from 2004 to 2017 were collected as samples, and a spatial econometric model was employed to un-dertake the empirical analysis. The specific research questions were: (1) How does land disposition derived from land finance influence urban innovation in China? Do spatial spillover effects exist? (2) What changes will take place in the degree of impact as the economy develops in China? (3) What are the similarities and differences in the impact, and how it changes, between developed regions and under-developed regions in China? As compared to the existing studies, the marginal contributions of this study can be summarized as follows: 1) The three aspects of land disposition, land finance and urban innovation have been integrated into a theoretical framework, which facilitates the systematic exploration of the relationship among them. 2) Both the positive and the negative impacts of land disposition on urban innovation have been discussed; bias may arise if one over-emphasizes the negative effects and neglects the positive. 3) In order to investigate the general features and regional heterogeneity in different areas, an em-pirical analysis has been undertaken at the national and regional levels, which is helpful in providing relevant implications and policy guidelines for governments that are based on the actual situation.

In the following sections, these questions will be fully analyzed and discussed. The rest of the paper is organized as follows. Section 2 presents the theoretical framework of this study. This is followed by a description of the data in section 3, and the establish-ment of the spatial econometric model in section 4. Section 5 reports the results and provides possible explanations for the findings. The important findings of the study are summarized, and their implications for policy advancement are discussed at the end. Page 1-3, line 34-147.

Point 3: Please higlight the novelty of presented study, it is totally omitted it in the current form.

Response 3: We are thankful to the respected reviewer for pointing out this problem. As advised, we have higlight the novelty of this study in the Introduction chapter.

As compared to the existing studies, the marginal contributions of this study can be summarized as follows: 1) The three aspects of land disposition, land finance and urban innovation have been integrated into a theoretical framework, which facilitates the systematic exploration of the relationship among them. 2) Both the positive and the negative impacts of land disposition on urban innovation have been discussed; bias may arise if one over-emphasizes the negative effects and neglects the positive. 3) In order to investigate the general features and regional heterogeneity in different areas, an em-pirical analysis has been undertaken at the national and regional levels, which is helpful in providing relevant implications and policy guidelines for governments that are based on the actual situation. Page 3, line 131-141.

Point 4: The quality of the figure 1 should be imporved, it is hard to read the text in the boxes.

Response 4: We are thankful to the respected reviewer for pointing out this problem. We apologize for the inconvenience caused. As advised, we have improved the quality of the figure 1 to present clearer text in the boxes. Page 5, line 219-220.

We really appreciate your friendly reminder. And we have already double check that all references are cited within the text, all citations within the text have a corresponding reference and the spelling of the authors’ names. Meanwhile the format of the reference list have also been unified according to the Guide for Authors.

Reviewer 3 Report

Congratulations to the authors of the article. In my opinion, the article has all the elements that a scientific paper requires.
The authors could make spatial maps to enrich the article, but this is not required for a full understanding of the work.
In my opinion, the selected journal does not correspond to the content of the article.

Author Response

Point 1: Congratulations to the authors of the article. In my opinion, the article has all the elements that a scientific paper requires.The authors could make spatial maps to enrich the article, but this is not required for a full understanding of the work.In my opinion, the selected journal does not correspond to the content of the article.

Response 1: We are thankful to the respected reviewer for pointing out this problem. As advised, we have present a spatial map to introduce the locations of 266 prefecture-level cities studied in this paper.

3.3. Data source

A city is a primary center of innovation activities. This study focuses on the influence of land dispositions on the innovation of prefecture-level cities in mainland China. Located in the east of Asia and the west coast of the Pacific Ocean, China has a land area of about 9.6 million km2 and consists of 333 prefecture-level cities. And there may distinguished differences in geographical positions, natural resource endowment, economic conditions, industrialization, urbanization among different cities. To ensure the integrity, continuity, and availability of the data, cities with missing values and outliers were omitted. Therefore, the panel data of 266 prefecture-level cities from 2004 to 2017 were collected into a sample to conduct empirical research. Page 10-11, line 459-4568.

Reviewer 4 Report

This study aims to deeply investigate the impacts of land disposition derived from land finance on urban innovation, incorporating the government-governing mechanism in China, to provide relevant implication and policy guideline for promoting urban innovation and optimizing land disposition in the context of implementing innovation-driven development strategy and deepening the reform and innovation comprehensively. The implementation of the tax-sharing reform and full open of land transfer market spawned land finance, intensified intergovernmental competition brought about by the GDP-oriented political promotion system and decentralization increased local government’s dependence on land-centered development, two land disposition modes gradually took shape and have underpinned rapid industrialization and urbanization in China over the past few decades. In order to understand whether such land disposition modes derived from land finance could continue to play an important role in the phase of high quality development driven by innovation, this study depicts detailed influence paths of what positive and negative effects LD mode1 and LD mode2 might exert on urban innovation in China and the theoretic mechanism is further investigated using the dynamic spatial Durbin model (DSDM) based on a panel data that covers 266 prefecture cities for the period 2004-2017

This is very good paper and should be published after minor revision as below.

General Remarks

Point 5 - Conclusions and Recommendations is too long. It should be reduced by 30-40%

Detailed remarks

  1. Point 3.2. The statistical methods used, their description and references should be added.
  2. Point 3.3. Some references should be added
  3. Figure 2. should be self-explaining. Please add proper information.
  4. Table 2, 3, 4, 5, 6, 7 should be self-explaining. Please add proper information.

Author Response

We are grateful for the time you have invested in the thoroughly assessment of our manuscript and for providing such valuable comments. Those comments are all valuable and very helpful for revising and improving our paper, as well as the important guiding significance to our researches. Moreover, we have studies comments carefully and have finished the corrections. We hope it meets with approval. The main corrections in the paper and the responds to your comments are as follows.

(We reply to the question highlighted with red color, the added context and significant change in revised manuscript highlighted with yellow in revised manuscript.)

Point 1: Point 5 - Conclusions and Recommendations is too long. It should be reduced by 30-40%.

Response 1: We are thankful to the respected reviewer for pointing out this problem. As advised, we have reorganized the Conclusions and Recommendations chapter. The whole chapter has been reduced by about 35% of the original version.

The implementation of the tax-sharing reform and the full opening of the land transfer market have given rise to land finance, and two land disposition modes have gradually taken shape in China. This study depicts in detail the paths of the positive and negative effects of LD mode1 and mode2 on urban innovation, and then further inves-tigates the theoretical mechanisms by employing a DSDM based on panel data covering 266 prefecture cities from 2004 to 2017.

The following key findings emerged: 1) The development of innovation was path-dependent, and the spatial agglomeration of urban innovation was confirmed. Meanwhile, the innovation performance of one city was found to inhibit the innovation levels of neighboring cities in the following year because of the Matthew effect. These phenomena were also evident in eastern areas, while the spillover effect was not ob-vious in mid-western cities. 2) In general, the combined impacts of land dispositions on urban innovation underwent a change from facilitative in the early stage to inhibitory at the current stage. The influence direction of LD mode1 on the urban innovation of local and neighboring cities was consistent, showing that positive impacts will be offset by negative impacts over time. The influence of LD mode2 followed a similar dynamic path, but here, the spillover effect on neighboring cities was shown to disappear in the long run. 3) Since the independent innovation capacities of eastern cities have already im-proved in the past few years, they depend less on government support to begin their innovation activities. Thus, the two LD modes could only help a little with urban in-novation, and the facilitative effect tends to disappear gradually. 4) The impact of land disposition on urban innovation in mid-western cities will change from promotional to inhibitory with time. Although the two LD modes can help underdeveloped cities ac-cumulate innovation funds and form industrial foundations in the early stage, the fa-cilitative effects would be offset by the fact that these areas are unappealing to innova-tion resources.

The following policy implications can be drawn from above findings: 1) The land transfer marketization reform should be moved forward to break the local government’s monopoly on land supply. The supply of different types of urban land is supposed to dynamically adjust itself, according to the real demands for urban development. 2) A rational structure for the supply of urban land should be provided to amend land dis-position modes derived from land finance. On the one hand, industrial land supply should be curtailed so that urban land use efficiency and the settlement threshold of traditional manufacturing enterprises can be improved. That said, residential and commercial land supply should be expanded to control the high land costs and soaring real estate prices, which will not only drive down urban living costs, but will also reduce the dependence of the local government on land finance. Meanwhile, a property tax ought to be introduced as a stable tax source. 3) The universal solution should be re-placed with distinct land disposition strategies according to the actual requirements of different regions, and the specific conditions and strategies for the innovation-driven development of different cities should be taken into consideration. 4) The performance evaluation system for local officials should be reformed. A GDP-oriented promotion system is not conducive to the improvement of cities with lower innovation capacities. Indexes related to urban innovation should be included, and economic indicators should be made less prominent to limit excessive inter-governmental competition.

This study has several limitations, which could be remedied in further research. The evaluation indicators of the two LD modes should be diversified; it would be helpful to carry out further in-depth analyses based on a comprehensive database, via which more diverse types of land transaction data could be included. Besides this, this study only reflects the effects of land disposition on urban innovation during relatively early stages, due to the unavailability of some indicators. It might make more sense to conduct further empirical analyses by incorporating more recent data, thus yielding resent a result with better goodness of fit. Page 20-21, line 770-821.

Point 2: Point 3.2. The statistical methods used, their description and references should be added.

Response 2: We are thankful to the respected reviewer for pointing out this problem. As advised, we have rearranged the Materials and Methods chapter, and the description of statistical methods employed have been added in 3.2. Empirical approach. This point mainly introduced the basic content Global Moran’s Index and Spatial Econometric Model. The cited references have also been added in the revised manuscript.

3.2. Empirical approach

3.2.1. Global Moran’s I Index

The innovation development of one city may show similar variation tendencies to neighboring cities due to the spillover and diffusion effects; such correlation between neighboring spatial units is called the spatial correlation [56]. Constructing a spatial econometric model is the main approach to investigating spatial correlation in some of the previous studies [8,13,30,57]. However, it is first necessary to examine whether the dependent variable has spatial dependence before conducting spatial economic analy-sis. The Global Moran’s I index is the most widely used method to measure whether a spatial correlation exists, and this reflects the degree of similarity between the attribute values of spatially adjacent or neighboring areas. The value is range [-1,1]. Greater than zero means the given variable shows positive spatial connections among its observa-tions and the existence of cluster in the spatial arrangement. Smaller than zero indicates that the given variable has negative spatial connection. When the value is zero, there is no spatial connection among the observation variables, and a random distribution in space can be inferred [58,59]. The calculation formula is as follows:

In Equation (1),  and  denote the urban innovation level of  city and  city respectively;  is the number of cities;  is the spatial weight matrix;  is the mean value of the samples; and  is the variance of the samples. In this study, a binary contiguity spatial weight matrix was constructed to reflect spatial adjacency. When region  is adjacent to region , =1; when region  is not adjacent to region , =0.

Page 9-10, line 396-415.

3.2.2. Spatial Econometric Model

According to the First Law of Geography, everything is related to everything, and closer things correlate more closely than distant ones; therefore, biased results may be generated if an economic model has not taken spatial analyses into account. A spatial economic model can address this problem by introducing spatial factors [60]. Besides this, the variable of the current time may be affected by the variable of the previous period because of a change in the time inertia of the variable, thus the variable’s spatial dependence may be reflected in both time and space. There are three basic spatial econometric models—the spatial lag model (SLM), the spatial error model (SEM), and the spatial Durbin model (SDM). The SLM only includes the lag term of the spatial dependence variable, and the SEM only includes the spatial spillover effect of the in-dependent variables, whereas the SDM includes both the lag term of the spatial de-pendence variable and the spatial spillover effects of the independent variables. Given that the SDM can be simplified to the SLM or the SEM under certain circumstances, this study incorporated the one-stage lag term of the explained variable into the dynamic spatial Durbin model (DSDM), in order to explore the influence of land dispositions on urban innovation. The specific DSDM can be expressed as follows [61]: Page 10, line 423-432.

It is necessary to identify which model suits the data best by conducting a series of model tests [62]. First, this study employed the Lagrange multiplier (LM) test and robust Lagrange multiplier test proposed by Anselin to test whether the model we established contains spatial interactions [63]. Meanwhile, the likelihood ratio (LR) test should be conducted to test whether the spatial panel model is more suitable for this study, and the SDM can be simplified to the SLM or the SEM. At the same time, the Hausman test should be employed to select the fixed effect and the random effect of the model. If the Hausman test is significant, the fixed effect model is adopted. If not, the random effect model should be selected. Page 10, line 449-457.

Point 3: Point 3.3. Some references should be added.

Response 3: We are thankful to the respected reviewer for pointing out this problem. To optimize the Materials and Methods chapter, we have rearrange this part according to the referees’ comments. Point 3.2.1. Global Moran’s Index introduces the basic content of the Moran´s I index and explained the reason it used in the revised manuscript. And we also added references cited in this part.

3.2.1. Global Moran’s I Index

The innovation development of one city may show similar variation tendencies to neighboring cities due to the spillover and diffusion effects; such correlation between neighboring spatial units is called the spatial correlation [56]. Constructing a spatial econometric model is the main approach to investigating spatial correlation in some of the previous studies [8,13,30,57]. However, it is first necessary to examine whether the dependent variable has spatial dependence before conducting spatial economic analy-sis. The Global Moran’s I index is the most widely used method to measure whether a spatial correlation exists, and this reflects the degree of similarity between the attribute values of spatially adjacent or neighboring areas. The value is range [-1,1]. Greater than zero means the given variable shows positive spatial connections among its observa-tions and the existence of cluster in the spatial arrangement. Smaller than zero indicates that the given variable has negative spatial connection. When the value is zero, there is no spatial connection among the observation variables, and a random distribution in space can be inferred [58,59]. The calculation formula is as follows:

In Equation (1),  and  denote the urban innovation level of  city and  city respectively;  is the number of cities;  is the spatial weight matrix;  is the mean value of the samples; and  is the variance of the samples. In this study, the binary contiguity spatial weight matrix was constructed to reflect the spatial adjacency. When region  is adjacent to region , =1; When region  is not adjacent to region , =0.

Page 9, line 381-401.

56 Odland, J. Spatial Autocorrelation. Sage Publication, Newbury Park, CA, USA, 1988. Page 23, line 938.

8 Fan, J.; Zhou, L. Three-dimensional intergovernmental competition, and urban sprawl: Evidence from Chinese prefec-tural-level cities. Land Use Policy 2019, 87, 104035. Page 21, line 848-849.

13 Wang, J.; Wu, Q.; Yan, S.; Guo, G.; Peng, S. China’s local governments breaking the land use planning quota: A strategic interaction perspective. Land Use Policy 2020, 92, 104434. Page 22, line 858-859.

30 Liu, Y. Government extraction and firm size: local officials’ responses to fiscal distress in China. J. Comp. Econ. 2018, 46, 1310-1331. Page 22, line 891-892.

57 Lu, X.; Wang, M.; Tang, Y. The spatial changes of transportation infrastructure and its threshold effects on urban land Use Efficiency: Evidence from China. Land. 2021, 10, 346. Page 23, line 939-940.

58 Moran P A P. A test for the serial independence of residuals. Biometrika. 1950, 37, 178-181. Page 23, line 941.

59 Bai, Y.; Deng, X.; Jiang, S.; Zhang, Q.; Wang, Z. Exploring the relationship between urbanization and urban eco-efficiency: Evidence from prefecture-level cities in China. J. Clean. Prod. 2018, 195, 1487-1496. Page 23, line 942-943.

Point 4: Figure 2. should be self-explaining. Please add proper information.

Response 4: We are thankful to the respected reviewer for pointing out this problem. As advised, we have further enriched the explanation of Figure 3 (which is Figure 2 in original manuscript) Moran’s I index of urban innovation from 2004 to 2016 in the revised manuscript. The contents are as follows.

4.1.1. Results of spatial Autocorrelation test

Since spatial correlation is one of the prerequisites for using a spatial economic model, the Global Moran’s I index was employed to verify whether spatial correlation exists in urban innovation among China’s prefecture-level cities. The variation trend in the Global Moran’s index of urban innovation can be seen in Figure. 3. The Moran’s I in-dexes of urban innovation were greater than zero, and the corresponding P values were statistically significant and positive at the 10% level, except in 2004. This means that the urban innovation levels of China’s prefecture-level cities are spatially dependent, and obvious spatial agglomeration can be observed. Therefore, a spatial econometric model is required to investigate the problems studied in this paper. Page 12, line 484-492.

Point 5: Table 2, 3, 4, 5, 6, 7 should be self-explaining. Please add proper information.

Response 5: We are thankful to the respected reviewer for pointing out this problem. As advised, we have further enriched the explanation of Table 2,3,4,5,6,7 in the revised manuscript. The contents are as follows.

Table 2 :

Table 2 shows the descriptive statistics of 266 prefecture-level administrative units in China from 2004 to 2017, including the observation of sample, mean value, standard deviation, minimum value and maximum value. Page 11, line 477-479.

Table 3 :

Table 3 reports the results of the LM test under space and time double-fixed effects, as well as the results of the LR test and the Hausman test. The results suggest that the LM test results regarding both the SLM and the SEM are significant at the 1% level. When using the robust LM test, a null hypothesis of no spatial lag effect was accepted at the 10% level, while a no spatial error effect was rejected, indicating that spatial dependence does exist in the data, and spatial panel models were more appropriate for the estima-tion. In the LR test, both null hypotheses: θ=0 and θ+ρβ=0 were significant at the 1% level, rejecting the assumption that SDM could be reduced to SLM and SEM, and demon-strating that the SDM is the most suitable for spatial panel data. Meanwhile, the results of Hausman test indicate that the assumption that the model has random effects can be rejected at the 1% significance level, indicating that the SDM with spatial fixed effects should be adopted in this study [64]. Page 12, line 497-509.

Table 4 :

As we can see from Table 4, the estimate of the one-stage time lag of urban innovation (INNOVt-1) was significantly positive at the 1% level in both models (2) and (4). The spatial lag coefficients of urban innovation (W×INNOV) were found to be statistically significant across different models, except for model (3). The spatial lag term (W×INNOVt-1) of urban innovation was significant positive in model (3), while it was significantly negative in model (4). Besides this, as a result of an integrated analysis of the estimated value of R-sq and Log-likelihood, as well as the economic logic implied by the estimated coefficient of the core explanatory variables, it is apparent that the DSDM under spatial and time-fixed effects had the optimal fitting specification in this study. Page 13, line 517-526.

Specifically, an average increase of 1% in urban innovation in the present year would drive a 1.307% increase in the next year. Page 13, line 533-534.

an average increase of 1% in innovation development in local city would lead to a 0.413% increase in neighboring cities at the same time. Page 13, line 538-540.

as an average increase of 1% in urban innovation in one city would lead to a 0.291% decrease in innovation in neighboring cities in the next year. Page 13, line 545-547.

Table 5 :

Therefore, we further decompose the total effects of the spatial model into direct effects and spatial spillover effects by means of the partial differentiation method. The results of this decomposition are summarized in Table 5. Page 15, line 596-598.

As for the control variables, both the direct and indirect effects of economic de-velopment level (INPGDP) on urban innovation were found to be statistically signifi-cant and positive in the short term, while the positive effects would become negative over time. Our results suggest that the promotional effect of economic development on local and neighboring cities’ urban innovation would disappear, and eventually change into an inhibitory effect with time. The direct and indirect effects of the scale of in-vestment in R&D (RDG), higher education level (EDU) and financial development level (FDL) on urban innovation show similar trends to the economic development level. One possible reason for this is that improvements in the local economy, the input into R&D, the higher education level, and the local financial situation might intensify the competition between neighboring governments, thereby giving rise to race to the bot-tom in innovation development in neighboring areas. In the long term, though, other pivot points of innovation development should be cultivated to enhance the innovation level in both local and neighboring cities. As for the degree of opening-up (OPE), the estimated direct effects were found to be statistically significant and negative in the short term, while this coefficient is insignificant in the long term, and the indirect effect shifts from significantly negative to positive with time. The results indicate that im-provements in local industries as a result of improving openness occur slowly, but may lead to intergovernmental competition over investment invitations, and may therefore promote innovation development in neighboring cities in the long term. Page 16, line 643-662.

Table 6 and 7 :

Specifically, an average increase of 1% in innovation level in the present year would lead to a 1.069% increase in the next year in East China, and a 1.222% increase in Midwest China. Page 17, line 680-682.

The results show that a 1% increase in urban innovation was associated with a 0.697% increase in innovation in neighboring cities in the East, while the estimate of the spatial spillover effect among midwestern cities is just 0.130%. Page 17, line 684-686.

In the East, an average increase of 1% in urban innovation in a local city in the present year would lead to a 1.564% decrease in innovation in neighboring cities in the next year, but the estimate of time and spatial lag in urban innovation in Midwest China was not statistically significant. Page 17, line 689-692.

The estimates of the direct and indirect effects of LTPD are 0.180 and 0.798 in East China, which are greater than those in Midwest China. This is also the case for the direct and indirect effects of PALA. Page 18, line 709-711.

Specifically, the estimates of the direct and indirect effects of LTPD and PALA were statistically non-significant, and the total effect continues to decrease with time in eastern cities. Meanwhile, the estimates of direct and indirect effects were all found to be statistically significant and negative in mid-western areas. Page 18, line 738-742.
